# RESIDUAL-GUIDED MULTI-RESOLUTION REFINEMENT OF FOUNDATION MODELS - A CASE STUDY IN CLIMATE FORECASTING

## ABSTRACT

Regional climate prediction presents unique challenges for time series foundation models, which typically process temporal patterns through a single-pass inference. Expert climatologists, in contrast, employ multi-scale temporal analysis and iterative refinement based on systematic error diagnosis. We present RGMR (Residual-Guided Multi-Resolution Refinement), an inference-time framework that adapts pre-trained foundation models to perform structured multi-scale reasoning for climate forecasting without parameter modification. Our approach combines hierarchical coarse-to-fine prediction refinement, and residual-guided error correction that systematically addresses prediction failures at each resolution level. Applied to drought forecasting using the Standardized Precipitation Evapotranspiration Index (SPEI), RGMR consistently enhances foundation model performance across diverse climate regions within an Australian regional area. Experimental results demonstrate substantial improvements over direct foundation model application, achieving up to 18.9% reduction in mean squared error, 10.2% reduction in root mean squared error, and 21.1% relative gain in $R^2$ when applied to TimesFM, with the largest benefits observed in climatologically complex regions where multi-scale temporal dynamics are most pronounced. The framework's inference-time operation enables immediate deployment on existing operational climate prediction systems without model retraining, offering a practical solution for enhancing foundation model capabilities in specialized forecasting domains.

## 1 INTRODUCTION

Climate prediction typically involves multi-stage analysis where meteorologists examine broad atmospheric patterns before progressively focusing on regional details, continuously adjusting forecasts as new evidence emerges (Lynch, 2006; Bauer et al., 2015). This iterative approach, where predictions evolve through structured analysis at multiple temporal scales, differs substantially from current AI systems that generate forecasts through single forward passes (Schultz et al., 2021).

Recent advances in time series foundation models (TSFMs) have demonstrated strong performance across diverse forecasting domains (Das et al., 2024; Marino et al., 2024; Gupta et al., 2024). However, TSFMs face challenges in regional climate prediction tasks such as drought forecasting, applications requiring nuanced understanding of local climate dynamics and multi-scale temporal patterns (Vicente Serrano et al., 2020; Mukherjee et al., 2018; Gao et al., 2022).

We hypothesize that this limitation stems from architectural differences in how these models process temporal information. Foundation models typically map historical sequences to future predictions in a single forward pass, processing all temporal information uniformly (Lim et al., 2021). Climatological analysis, by contrast, often involves iterative examination of data at multiple time scales (Trenberth et al., 2009), identification of systematic patterns and biases (Jolliffe & Stephenson, 2003), and progressive refinement through focused analysis of specific temporal components (Palmer, 2000).

Recent work in iterative refinement for sequential prediction tasks (Wei et al., 2022; Kojima et al., 2022; Yao et al., 2023) suggests that multi-stage processing can improve performance on complex reasoning tasks. This motivates our central question: *Can time series foundation models benefit from structured multi-stage reasoning approaches for climate forecasting?*

Figure 1 illustrates the difference between multi-scale temporal analysis and single-pass processing using SPEI time series data. Climatological analysis typically decomposes multi-scale climate patterns into constituent temporal components (annual cycles, ENSO oscillations, and extreme events), enabling systematic examination of each scale before integration. Current AI models process the entire complex signal simultaneously, which may contribute to systematic prediction errors during drought events and regime shifts.

The analogy between iterative context refinement in language tasks and error-corrective reasoning in climate prediction suggests several shared requirements: (1) forming initial hypotheses based on available evidence, (2) evaluating these hypotheses, (3) identifying specific weaknesses, and (4) iteratively refining understanding through targeted analysis (Shinn et al., 2023; Madaan et al., 2023).

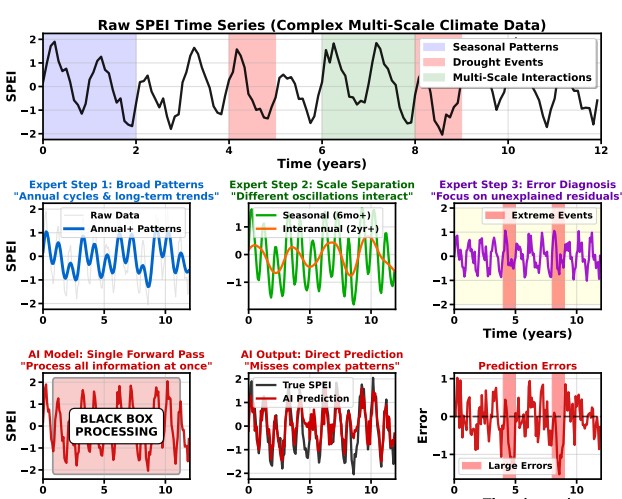

However, adapting iterative refinement to climate forecasting presents unique challenges. Unlike language tasks where reasoning steps can be explicitly verbalized (Wei et al., 2022), climate analysis must operate through learned representations in foundation models. Additionally, operational climate systems require efficient inference, making parameter modification of foundation model approaches impractical (Bauer et al., 2015).

Figure 1: Comparison of temporal processing approaches in climate forecasting. Multi-scale decomposition analysis (middle row) versus single forward pass processing (bottom row) for complex climate pattern prediction.

To address these challenges, we propose **Residual-Guided Multi-Resolution Refinement (RGMR)**, a framework that enables time series foundation models to perform structured multi-scale reasoning without modifying any parameters of the backbone foundation model. Our approach uses background knowledge to confirm relevant temporal scales, then iteratively refines predictions from coarse to fine resolutions through residual-guided error correction.

We evaluate RGMR on drought forecasting using SPEI (Vicente-Serrano et al., 2010) data across multiple South Australian locations. Results show consistent improvements over direct foundation model application, with up to 18.9% error reduction in mean squared error.

**Our main contributions include:**

- We investigate the temporal processing differences between current AI forecasting approaches and multi-scale climatological analysis, proposing multi-resolution refinement as a potential solution.
- We provide empirical analysis of information preservation in multi-resolution processing and demonstrate convergence properties of residual-guided refinement, with background knowledge foundations for scale selection.
- We demonstrate that multi-scale reasoning approaches can substantially improve regional climate forecasting accuracy.

## 2 RELATED WORK

**Drought forecasting.** Classical statistical models (e.g., ARIMA, SVMs) have long been used for drought indices (Vicente-Serrano et al., 2015; Nandgude et al., 2023) but depend on bespoke feature engineering and struggle with long-range dependencies. Deep architectures—including LSTMs (Hochreiter & Schmidhuber, 1997), Transformer-based forecasters (Vaswani et al., 2017; Zhou

et al., 2021), and recent spatiotemporal models (Tan et al., 2025) improve expressivity but typically require task-specific training or fine-tuning to a region and target index. In contrast, we study an *inference-time* framework that adapts a frozen base model to drought forecasting without updating weights.

**Time-series foundation models (TSFMs).** General-purpose TSFMs such as TimesFM (Das et al., 2024), TimeGPT-1 (Garza & Mergenthaler-Canseco, 2023), Lag-Llama (Marino et al., 2024), Chronos (Gupta et al., 2024), and Timer (Li et al., 2024) aim for broad-domain forecasting with a single pre-trained backbone. Domain-specialized weather systems like GraphCast (Lam et al., 2023) and Pangu-Weather (Bi et al., 2023) achieve strong numerical weather prediction but are not designed as drop-in forecasters for regional drought indices and usually involve re-training or domain-specific pipelines. We target the complementary setting of *using* a general TSFM as-is and improving its outputs at inference time.

**Multi-resolution and residual learning.** Multi-scale techniques capture dynamics across temporal scales, e.g., Scaleformer's coarse-to-fine pathway (Shabani et al., 2023), N-BEATS' hierarchical residual stacks (Oreshkin et al., 2019), and Minusformer's progressive residual refinement (Liang et al., 2024). These methods modify model architectures or training procedures to realize multi-resolution behavior. Our approach differs in *where* multi-resolution acts: we implement coarse-to-fine *refinement at inference time* on top of a frozen backbone, using residuals learned on short windows to correct long-window proposals without re-training the base.

**Inference-time adaptation.** Test-time adaptation (Sun et al., 2020) and lightweight prompting/tuning (Brown et al., 2020) enable on-the-fly specialization while retaining a pre-trained model's generality. For time series, few-shot or episodic adaptation has been explored mostly via fine-tuning (Oreshkin et al., 2021). Our RGMR framework performs *inference-time* adaptation: it composes multi-resolution proposals and residual corrections with backtracking-style gating, preserving the pre-trained TSFM and avoiding any weight updates.

*Summary.* Relative to prior drought models (training-time), multi-resolution/residual architectures (training-time), and test-time adaptation (often with updates), RGMR occupies a distinct point in the design space: a frozen-TSFM, multi-resolution, residual-guided *inference-time framework* that requires no re-training and is thus readily deployable.

## 3 PROBLEM DEFINITION

We study the task of forecasting the Standardized Precipitation–Evapotranspiration Index (SPEI), a widely used drought index that integrates precipitation and potential evapotranspiration into a single standardized time series. Let $\{y_t\}_{t=1}^T$ denote the monthly SPEI values at a specific region, and $\{\mathbf{X}_t\}_{t=1}^T$ the associated $D$-dimensional covariate sequence from climate reanalysis data. All preprocessing and normalization statistics are computed only on the training split.

**Multi-step forecasting.** We consider forecasting the next $H$ steps. At forecast origin $t$, the goal is to produce a vector prediction $\hat{\mathbf{y}}_{t+1:t+H} \in \mathbb{R}^H$ for the next $H$ SPEI values $\mathbf{y}_{t+1:t+H}$. A frozen foundation model $f_\theta$ maps a length-$W$ context window to an $H$-step forecast:

$$\hat{\mathbf{y}}_{t+1:t+H} = f_\theta\big(\mathbf{X}_{t-W+1:t}\big), \qquad f_\theta : \mathbb{R}^{W \times D} \to \mathbb{R}^H,$$

where $\mathbf{X}_{t-W+1:t} \in \mathbb{R}^{W \times D}$ denotes the multivariate input context. The one-step case is recovered by setting $H = 1$. Building on this formulation, we develop an inference-time framework that systematically improves the base prediction quality. The notation used throughout is summarized in Table 1.

## 4 RESIDUAL-GUIDED MULTI-RESOLUTION REFINEMENT (RGMR)

### 4.1 METHOD OVERVIEW

RGMR is an inference-time framework that adapts *frozen* time-series foundation models through structured multi-scale reasoning. We fix a resolution hierarchy $\mathcal{R} = \{r_1 = 12, r_2 = 6, r_3 = 3, r_4 = 2, r_5 = 1\}$ in months from coarse to fine. For each $r_k$, the frozen backbone produces a full-context

Table 1: Symbols used in the main text (scalars roman; vectors bold; sets/operators calligraphic). We focus on $H{=}1$ in the main results; for $H{>}1$ the scalar quantities generalize elementwise to $\mathbb{R}^H$.

| Symbol | Meaning |
|---|---|
| $W$ | full inference window length passed to $f_\theta$ |
| $L_{\text{short}}$ | short training window for residual learning (e.g., 12 months) |
| $\mathcal{R} = \{r_1 > \cdots > r_K\}$ | resolution (stride) ladder from coarse to fine; $K{=}\|\mathcal{R}\|$ |
| $\mathcal{D}_r, \mathcal{U}_r$ | down-/up-projection at stride $r$ (block average; repeat) |
| $\mathcal{P}_r = \mathcal{U}_r \circ \mathcal{D}_r$ | projection operator at resolution $r$ |
| $C^{(k)}$ | level-$k$ base prediction from $f_\theta$ at resolution $r_k$ (inference) |
| $\hat{y}^{(k)}$ | refined prediction after level $k$ (initialize $\hat{y}^{(1)}{=}C^{(1)}$) |
| $\tilde{y}^{(k)}$ | *short-window* base prediction used to form training residuals |
| $\tilde{R}^{(k)}$ | *training* residual at level $k$: $\tilde{R}^{(k)} = y - \tilde{y}^{(k)}$ |
| $\widehat{\tilde{R}}^{(k)}$ | predicted training residual (from Ridge $g_\phi^{(k)}$) used at inference |
| $R^{(k)}$ | *inference* residual at level $k$: $R^{(k)} = y - \hat{y}^{(k)}$ |
| $\mathbf{w}^{(k)}$ | adaptive weight vector at level $k$ for residual correction |
| $\mathbf{w}_{t,i}^{(k)}$ | $i$-th component of $\mathbf{w}^{(k)}$ at time $t$ (elementwise weighting) |
| $\alpha^{(k)}$ | mixing coefficient at level $k$ (monotone, coarse$\rightarrow$fine increasing) |
| $\eta^{(k)}$ | adaptive step size at level $k$ (with backtracking, $\eta^{(k)} \in (0, 2]$) |
| $\rho^{(k)}$ | contraction factor at level $k$ (defined in Theorem 1) |
| $B_k$ | bounded error term at level $k$, from proposal error and residual-prediction noise |

proposal. During training and validation only, we fit lightweight per-scale residual predictors on short-window inputs to capture systematic, scale-specific errors. At test time, no ground truth is used and no backbone weights are updated. Proposals are refined from coarse to fine by applying the learned residual predictors to the long-window proposals, with clipped weights and an optional backtracking line search to ensure stability. This two-phase design learns residual error patterns on short contexts and then corrects long-context multi-resolution proposals at inference, targeting failure modes of single-pass inference such as phase misalignment and regime shifts while keeping the foundation model unchanged. Sections 4.1 to 4.5 detail the projections and ladder (4.1), per-scale proposals (4.2), residual learning and features (4.3), the inference-time update rule (4.4), and a contraction-style analysis explaining why errors decay down to a noise floor (4.5).

## 4.2 MULTI-RESOLUTION PROJECTION OPERATIONS

For each resolution level $r \in \mathcal{R}$, we apply temporal projection operations

$$\mathcal{P}_r = \mathcal{U}_r \circ \mathcal{D}_r, \tag{1}$$

where $\mathcal{D}_r$ downsamples by averaging within non-overlapping length-$r$ segments and $\mathcal{U}_r$ upsamples by repeating values. Using zero-based indexing (so $ir = i \times r$) and letting $\lfloor \cdot \rfloor$ denote the floor operator, we have $\mathcal{D}_r(x)[i] = \frac{1}{r} \sum_{j=ir}^{(i+1)r-1} x[j]$ and $\mathcal{U}_r(\mathcal{D}_r(x))[j] = \mathcal{D}_r(x)[\lfloor j/r \rfloor]$. This yields multi-resolution temporal views of the input while preserving information and maintaining computational efficiency.

## 4.3 RESIDUAL PATTERN LEARNING

To train residual predictors, we first generate predictions using short context windows during the training and validation phases. For each forecast origin $t$ in these phases and each resolution level $k$, we apply the foundation model with a reduced window length $L_{short}$ (typically 12 months):

$$\tilde{y}_{t+1}^{(k)} = f_\theta(\mathcal{P}_{r_k}(\mathbf{X}_{t-L_{short}+1:t})) \tag{2}$$

We then compute residuals $\tilde{R}_{t+1}^{(k)} = y_{t+1} - \tilde{y}_{t+1}^{(k)}$, which capture systematic error patterns at each resolution level when using limited context.

Using these collected residuals, we train Ridge regression models to predict future residuals. For each level $k$, we construct feature vectors $\mathbf{z}_t$ containing recent target values, rolling statistics (mean, standard deviation, linear trend), recent residual history, and the resolution level index. The Ridge regressor is formulated as $g_\phi^{(k)}(\mathbf{z}_t) = \mathbf{w}^{(k)\top}\mathbf{z}_t$, trained by minimizing $\sum_{t \in \text{Train}}(\tilde{R}_{t+1}^{(k)} - \mathbf{w}^{(k)\top}\mathbf{z}_t)^2 + \lambda^{(k)}\|\mathbf{w}^{(k)}\|_2^2$ with $\lambda^{(k)}$ selected via time ordered cross validation.

### 4.4 INFERENCE-TIME ITERATIVE REFINEMENT

At test time, RGMR uses the full context window $W$ (70% of the series length). Let the resolution ladder be $\mathcal{R} = \{r_1 > \cdots > r_K\}$ with $r_1$ the coarsest. We initialize

$$\hat{\mathbf{y}}^{(1)}_{t+1:t+H} = f_\theta\big(\mathcal{P}_{r_1}(\mathbf{X}_{t-W+1:t})\big). \tag{3}$$

For each subsequent level $k = 2, \dots, K$, we query the frozen backbone at resolution $r_k$ to obtain a level-$k$ base forecast, $\mathbf{C}^{(k)}_{t+1:t+H} = f_\theta\big(\mathcal{P}_{r_k}(\mathbf{X}_{t-W+1:t})\big)$, and predict the short-window residual $\widehat{\mathbf{R}}^{(k-1)}_{t+1:t+H} = g^{(k-1)}_\phi(\mathbf{z}_t)$. We form adaptive element-wise weights

$$\mathbf{w}^{(k-1)}_{t+1:t+H} = \text{clip}_{[\varepsilon_{\min}, \, 1]}\left(\sigma\left(\gamma\left(\left|\widehat{\mathbf{R}}^{(k-1)}_{t+1:t+H}\right| - \delta^{(k-1)}\right)\right)\right), \tag{4}$$

with $\sigma(x) = 1/(1+e^{-x})$, where $\gamma > 0$ is a small slope constant, $\delta^{(k)}$ is a validation-selected quantile of $\left|\widehat{\mathbf{R}}^{(k)}\right|$, and $\varepsilon_{\min}$ is a stability floor (details in App. D.3). The mixing coefficient $\alpha^{(k)}$ increases with resolution (e.g., $\alpha^{(k)} = 0.3 + 0.5(1 - r_k/\max(\mathcal{R}))$). The step size $\eta^{(k)} \in (0, 2]$ is chosen by backtracking to ensure a non-expansive correction update (see App. E for the precise condition and proof). We then update

$$\hat{\mathbf{y}}^{(k)}_{t+1:t+H} = \alpha^{(k)} \mathbf{C}^{(k)}_{t+1:t+H} + \big(1 - \alpha^{(k)}\big)\left(\hat{\mathbf{y}}^{(k-1)}_{t+1:t+H} + \eta^{(k)} \mathbf{w}^{(k-1)}_{t+1:t+H} \odot \widehat{\mathbf{R}}^{(k-1)}_{t+1:t+H}\right). \tag{5}$$

This progressively corrects errors while preserving the foundation model's multi-scale reasoning.

### 4.5 THEORETICAL PROPERTIES

We next analyze why RGMR improves across refinement levels. Each residual-guided update shrinks the expected error by a factor $\rho^{(k)} < 1$ while adding only bounded noise from residual prediction, yielding geometric decay to a noise floor. Building on the contraction-mapping view (and fixed-point formulations such as DEQ) but differing in composing level-specific contractions and explicitly modeling bounded stochastic residual noise (Granas et al., 2003; Bai et al., 2019), we state the following result.

**Theorem 1 (Residual-guided contraction)** *Define the contraction factor*

$$\rho^{(k)} = (1 - \alpha^{(k)}) \max_i \left|1 - \eta^{(k)} \mathbf{w}^{(k-1)}_{t,i}\right|. \tag{6}$$

*If $\alpha^{(k)} \in (0, 1)$, $\eta^{(k)} \in (0, 2]$, and $\mathbf{w}^{(k-1)}_{t,i} \in [0, 1]$, then $\rho^{(k)} < 1$ and*

$$\mathbb{E}\left[\|y_t - \hat{y}^{(k)}_t\|_2^2\right] \leq (1 + \alpha^{(k)})(\rho^{(k)})^2 \, \mathbb{E}\left[\|y_t - \hat{y}^{(k-1)}_t\|_2^2\right] + B_k, \tag{7}$$

*where $B_k$ collects bounded contributions from the level-$k$ proposal error and residual-prediction noise. Moreover, since $\rho^{(k)} \leq 1 - \alpha^{(k)}$, the leading factor satisfies $(1 + \alpha^{(k)})(\rho^{(k)})^2 \leq (1 + \alpha^{(k)})(1 - \alpha^{(k)})^2 < 1$, ensuring a strict contraction at each level. (An explicit upper bound for $B_k$ is provided in the appendix.)*

*Proof sketch.* Substitute the RGMR update and bound the spectral norm of $(I - \eta^{(k)}\text{Diag}(\mathbf{w}^{(k-1)}_t))$ (diagonal with entries in $[0, 1]$); together with $\alpha^{(k)} > 0$ this yields $\rho^{(k)} < 1$. Apply norm submultiplicativity and two applications of Young's inequality to obtain the second-moment bound.

**Corollary 1 (Finest-level comparison)** *Let the direct finest-resolution baseline be $\hat{\mathbf{y}}_{\text{dir}} = \mathbf{C}^{(K)} = f_\theta(\mathcal{P}_{r_K}(\mathbf{X}_{t-W+1:t}))$, which involves no residual correction, weights, or step sizes. Since $\|\mathbf{y}^\star - \mathbf{C}^{(K)}\|_\infty \leq B_K$, whenever*

$$\max_{j \in [H]}\left|1 - \eta^{(K)} w^{(K-1)}_{t+1:t+H, j}\right| \|\mathbf{y}^\star - \hat{\mathbf{y}}^{(K-1)}\|_\infty + \eta^{(K)}\|\mathbf{w}^{(K-1)}\|_\infty E_{K-1} \leq B_K,$$

*it holds that*

$$\|\mathbf{y}^\star - \hat{\mathbf{y}}^{(K)}\|_\infty \leq \|\mathbf{y}^\star - \hat{\mathbf{y}}_{\text{dir}}\|_\infty.$$

### 4.6 ALGORITHM DESCRIPTION

To operationalize RGMR, we split the procedure into two phases: (1) *training* lightweight residual predictors using short-window contexts, and (2) *inference-time refinement* over multiple temporal resolutions with the frozen foundation model. Algorithm 1 summarizes the training stage, where residual predictors are learned from systematic errors identified on SPEI sequences.

---

**Algorithm 1** RGMR Training: Residual Predictor Learning

---

**Require:** Training series $\{(\mathbf{X}_s, y_s)\}_{s=1}^{T_{\text{train}}}$, frozen model $f_\theta$, short window $L_{\text{short}}$, ladder $\mathcal{R} = \{r_1 = 12, r_2 = 6, r_3 = 3, r_4 = 2, r_5 = 1\}$, horizon $H$

1: **Initialize:** $\mathcal{D}^{(k)} \leftarrow \emptyset$ for all $k \in \{1, \ldots, 5\}$        $\triangleright$ Level-wise residual datasets

2: **for** each forecast origin $t$ with $t + H \leq T_{\text{train}}$ **do**

3:      **for** each level $k \in \{1, \ldots, 5\}$ **do**

4:          **Short-window base:** $\tilde{\mathbf{y}}_{t+1:t+H}^{(k)} \leftarrow f_\theta\big(\mathcal{P}_{r_k}(\mathbf{X}_{t-L_{\text{short}}+1:t})\big)$

5:          **Training residual:** $\tilde{\mathbf{R}}_{t+1:t+H}^{(k)} \leftarrow \mathbf{y}_{t+1:t+H} - \tilde{\mathbf{y}}_{t+1:t+H}^{(k)}$

6:          **Features:** $\mathbf{z}_t^{(k)} \leftarrow$ [recent targets, rolling stats, residual history, level $k$]

7:          **Append:** $\mathcal{D}^{(k)} \leftarrow \mathcal{D}^{(k)} \cup \big\{(\mathbf{z}_t^{(k)}, \tilde{\mathbf{R}}_{t+1:t+H}^{(k)})\big\}$

8: **for** each level $k \in \{1, \ldots, 5\}$ **do**

9:      **Train multi-output Ridge:** $g_\phi^{(k)} \leftarrow \arg\min_{\mathbf{B} \in \mathbb{R}^{p \times H}} \sum_{(\mathbf{z}, \mathbf{r}) \in \mathcal{D}^{(k)}} \big\| \mathbf{r} - \mathbf{B}^\top \mathbf{z} \big\|_2^2 + \lambda^{(k)} \|\mathbf{B}\|_F^2$

     $\triangleright$ $p = \dim(\mathbf{z})$

10: **Return** residual predictors $\{g_\phi^{(k)}\}_{k=1}^5$

---

Having obtained residual predictors, we apply them at inference time to refine multi-resolution forecasts generated by the foundation model. Algorithm 2 describes this process: starting from the coarsest resolution, RGMR progressively integrates finer-scale predictions, uses residual forecasts for self-critique, and enforces contraction conditions to guarantee stable error reduction across refinement levels.

---

**Algorithm 2** RGMR Inference: Multi-Resolution Refinement

---

**Require:** Context $\mathbf{X}_{t-W+1:t}$, frozen model $f_\theta$, residual predictors $\{g_\phi^{(k)}\}_{k=1}^5$, ladder $\mathcal{R} = \{r_1, \ldots, r_5\}$ with $r_1 = 12$, horizon $H$

1: **Init (coarsest):** $\hat{\mathbf{y}}_{t+1:t+H}^{(1)} \leftarrow f_\theta\big(\mathcal{P}_{r_1}(\mathbf{X}_{t-W+1:t})\big)$

2: **for** $k = 2$ to $5$ **do**

3:      **Perceive (level $k$ base):** $\mathbf{C}_{t+1:t+H}^{(k)} \leftarrow f_\theta\big(\mathcal{P}_{r_k}(\mathbf{X}_{t-W+1:t})\big)$

4:      **Build z:** $\mathbf{z}_t^{(k-1)} \leftarrow \text{FeatExtract}\Big(\mathcal{P}_{r_{k-1}}(\mathbf{X}_{t-L_{\text{short}}+1:t}), \hat{\mathbf{y}}_{t+1:t+H}^{(k-1)}, \mathbf{C}_{t+1:t+H}^{(k)}, r_{k-1}, t\Big)$

5:      **Residual Prediction:** $\widehat{\tilde{\mathbf{R}}}_{t+1:t+H}^{(k-1)} \leftarrow g_\phi^{(k-1)}(\mathbf{z}_t^{(k-1)})$

6:      **Weigh (elementwise):** $\mathbf{w}_{t+1:t+H}^{(k-1)} \leftarrow \text{clip}_{[\varepsilon_{\min}, 1]}\Big(\sigma\big(\gamma\big(\big|\widehat{\tilde{\mathbf{R}}}_{t+1:t+H}^{(k-1)}\big| - \delta^{(k-1)}\big)\big)\Big)$

7:      **Choose step (backtracking):** pick $\eta^{(k)} \in (0, 2]$ s.t. $\rho^{(k)} = (1 - \alpha^{(k)}) \max_{j \in [H]} \big|1 - \eta^{(k)} w_{t+1:t+H, j}^{(k-1)}\big| < 1$

8:      **Act (refine):** $\hat{\mathbf{y}}_{t+1:t+H}^{(k)} \leftarrow \alpha^{(k)} \mathbf{C}_{t+1:t+H}^{(k)} + (1 - \alpha^{(k)})\Big(\hat{\mathbf{y}}_{t+1:t+H}^{(k-1)} + \eta^{(k)} \mathbf{w}_{t+1:t+H}^{(k-1)} \odot \widehat{\tilde{\mathbf{R}}}_{t+1:t+H}^{(k-1)}\Big)$

9: **Return** $\hat{\mathbf{y}}_{t+1:t+H}^{(5)}$

---

Together, Algorithms 1 and 2 instantiate the RGMR workflow: residual predictors capture systematic biases in SPEI forecasting, while multi-resolution refinement ensures consistent geometric error reduction at inference time without retraining the base model.

### 4.7 COMPUTATIONAL COMPLEXITY AND IMPLEMENTATION

Let $K = |\mathcal{R}|$ denote the number of resolution levels (in our setup $K = 5$). Per refinement cycle, RGMR performs $K$ calls to the frozen foundation model $f_\theta$ (one per level), $K-1$ residual predictions using a lightweight Ridge regressor, and projection operations whose total cost is linear in the context length. The overall inference complexity is

$$\mathcal{O}\big(K \cdot C_\theta \ + \ K \cdot W \ + \ (K-1) \cdot C_{\text{ridge}}^{\text{pred}}\big),$$

where $C_\theta$ is the cost of a single $f_\theta$ forward pass, $W$ is the input window length, and $C_{\text{ridge}}^{\text{pred}} = \mathcal{O}(p+q)$ is the per-step cost of residual feature evaluation (e.g., $p$ lags and a $q$-step trend) plus a closed-form Ridge prediction. In practice $C_{\text{ridge}}^{\text{pred}} \ll C_\theta$, so the end-to-end runtime (elapsed time) is dominated by $K \cdot C_\theta$. The backtracking line search at each level performs at most $J_{\max} = \lceil \log_2(\eta_{\text{init}}/\eta_{\min}) \rceil$ halvings, hence adds only a $\mathcal{O}(K \cdot J_{\max})$ factor of lightweight element-wise operations, which is negligible relative to $f_\theta$ calls.

RGMR operates entirely at inference time: there are *no parameter updates* to $f_\theta$, enabling immediate deployment on existing time-series foundation models without retraining. Hyperparameters are kept minimal and use conservative defaults (e.g., fixed ladder $\mathcal{R}$, weight-floor $\varepsilon_{\min}$, and the backtracking envelope $\eta \in (0, 2]$). When selection is required (e.g., residual-weight threshold quantiles $\delta^{(k)}$ or the Ridge penalty $\lambda_{\text{ridge}}$), we employ a *single* time-ordered validation pass; no test-time peeking or re-training of $f_\theta$ is involved. The adaptive weighting mechanism is implemented via element-wise sigmoids and simple clipping, and thus contributes negligible overhead compared to the $K$ forward calls of $f_\theta$.

## 5 EXPERIMENTS

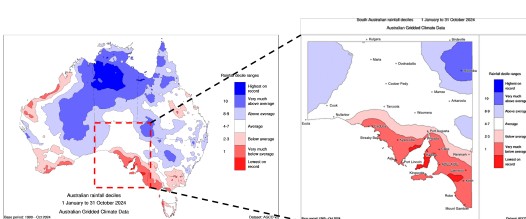

Figure 2: Study Area

We evaluate whether RGMR improves frozen foundation models for S2S SPEI forecasting without parameter updates. The study targets overall accuracy against strong baselines, the effect of multi-resolution and residual refinement, the role of residual-guided refinement with backtracking, and computational overhead. We use rolling-origin evaluation with non-overlapping test windows (step size $s = H$); all standardization uses training-set statistics; the residual predictor is trained on the training split; test targets are never accessed at inference. Datasets, regions, covariates and splits are summarized in Appendix C; implementation details and hyperparameters are in Appendix D. Code is available at `https://anonymous.4open.science/r/RGMR_implementation-F30D`

### 5.1 SETUP

We evaluate regional SPEI forecasting across multiple nominal climate zones within South Australia (arid, temperate) and decades of records to ensure geographic and climatic diversity; exact regions, time spans, and sample counts are reported in Appendix C. Targets are SPEI at horizons $H$ months ahead; inputs are multivariate climate covariates and past SPEI within a window of length $W$, shared across all methods.

**Study area.** We focus on South Australia (as shown in Figure 2) as the study area for two main reasons. First, the region is highly vulnerable to drought and interannual rainfall variability, making seasonal prediction societally important. Second, reliable meteorological records (NCEP NCAR Reanalysis data) are available for this region, ensuring reproducibility. Recent reports show that parts of South Australia recorded their driest year on record in 2024, with rainfall totals at multiple locations reaching historic lows between June 2024 and May 2025 (ABC News, 2025b;a). Such severe and prolonged droughts underline the acute need for accurate climate forecasting in the region.

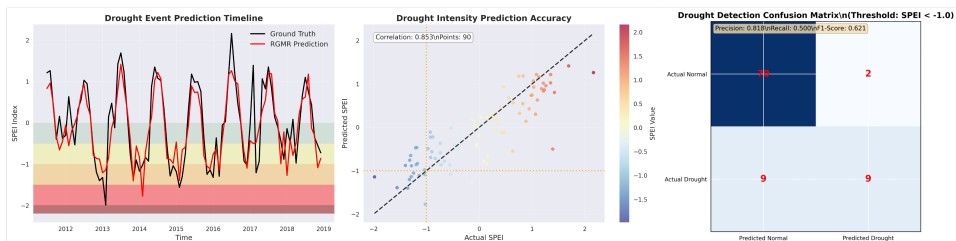

Figure 3: RGMR forecast quality and drought-event detection at a representative South Australia site. Left: monthly SPEI timeline with ground truth (black) and RGMR prediction (red); shaded bands indicate SPEI categories from severe drought to wet. Middle: predicted vs. actual SPEI with $y=x$ (dashed); Pearson $r=0.83$ over $n=90$ test months; point color encodes SPEI. Right: confusion matrix for drought detection at threshold SPEI $\leq -1.0$ (precision 0.818, recall 0.500, F1 0.621); cell values are counts and colors reflect relative frequency.

**Baselines and protocol.** We compare against frozen foundation models and strong multi-resolution and statistical baselines. *Foundation-only* includes TimesFM, TimeGPT, and TabPFN on the raw window; we also report an equal-weight ensemble of these foundation predictions. *Multi-resolution* baselines include N-BEATS (5 stacks with standard trend/seasonality blocks), Scaleformer with learned scale selection, and a fixed ladder $\{12, 6, 3, 2, 1\}$. We include a *residual-refinement* variant atop the foundation-only model *without* projections. All methods use identical inputs, splits, and preprocessing; none updates $f_\theta$ during inference. We report MSE, MAE, RMSE, and $R^2$ per region, and aggregate by median and interquartile range.

## 5.2 MAIN RESULTS

### 5.2.1 ONE-MONTH-AHEAD SPEI

Figure 3 provides an overview of RGMR performance on South Australian SPEI, illustrating both forecasting quality and downstream drought-event detection. Figure 4 illustrates the progressive reduction in error across RGMR refinement levels on representative cases, consistent with the residual-guided contraction guarantee in Section 4. Table 2 reports one-month-ahead results at three South Australian locations, including N-BEATS and Scaleformer as multi-resolution baselines.

Foundation models (TabPFN, TimeGPT, TimesFM) already outperform generic deep baselines, and applying RGMR on top of TimesFM yields the strongest performance across all metrics and sites. Specifically, MSE decreases from 0.391 to **0.318** at Location 1, from 0.318 to **0.258** at Location 2, and from 0.524 to **0.426** at Location 3, with corresponding RMSE reductions (e.g., 0.625→**0.561** at Location 1). $R^2$ improves consistently, reaching up to 0.568 at Location 3.

Compared directly with multi-resolution baselines, TimesFM+RGMR achieves lower MSE than both Scaleformer and N-BEATS at all sites (e.g., Location 1: 0.318 vs 0.472/0.482; Location 2: 0.258 vs 0.505/0.512; Location 3: 0.426 vs 0.662/0.668), with similar gaps across other metrics. These results highlight RGMR's ability to deliver consistent gains beyond both generic deep baselines and established multi-resolution architectures.

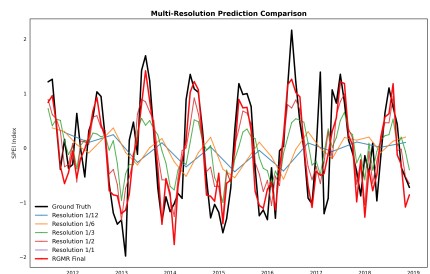

Figure 4: Multi-resolution refinement on a representative case: per-level error decreases monotonically as RGMR refines predictions.

Table 3 averages results over the three locations and decomposes RGMR into its components. Using a single coarse resolution improves the foundation baselines, multi-resolution without residual refinement brings further gains, residual-guided updates add monotonic improvements, and adaptive residual weights attain the best overall performance. This progression validates our design choices: the per-level update achieves contraction with modulus $\rho^{(k)} < 1$ under the stability conditions

Table 2: One-month-ahead SPEI forecasting at three South Australian locations. Best results in **bold**.

| Method | Location 1 (-26.125_129.125) | | | | Location 2 (-29.125_134.875) | | | | Location 3 (-35.625_138.875) | | | |
|---|---|---|---|---|---|---|---|---|---|---|---|---|
| | MSE↓ | MAE↓ | RMSE↓ | R²↑ | MSE↓ | MAE↓ | RMSE↓ | R²↑ | MSE↓ | MAE↓ | RMSE↓ | R²↑ |
| *Multi-resolution baselines* | | | | | | | | | | | | |
| N–BEATS (5 stacks) | 0.482 | 0.497 | 0.658 | 0.505 | 0.512 | 0.517 | 0.680 | 0.465 | 0.668 | 0.590 | 0.777 | 0.409 |
| Scaleformer | 0.472 | 0.491 | 0.652 | 0.510 | 0.505 | 0.511 | 0.674 | 0.469 | 0.662 | 0.585 | 0.771 | 0.413 |
| *Generic deep baselines* | | | | | | | | | | | | |
| Transformer | 0.537 | 0.554 | 0.733 | 0.451 | 0.573 | 0.575 | 0.757 | 0.414 | 0.745 | 0.655 | 0.863 | 0.365 |
| Autoformer | 0.546 | 0.559 | 0.739 | 0.442 | 0.582 | 0.580 | 0.763 | 0.405 | 0.752 | 0.658 | 0.867 | 0.358 |
| Crossformer | 0.542 | 0.557 | 0.736 | 0.446 | 0.578 | 0.578 | 0.760 | 0.409 | 0.740 | 0.654 | 0.860 | 0.370 |
| DLinear | 0.533 | 0.551 | 0.730 | 0.455 | 0.569 | 0.572 | 0.754 | 0.418 | 0.735 | 0.650 | 0.857 | 0.375 |
| FiLM | 0.558 | 0.569 | 0.747 | 0.430 | 0.594 | 0.590 | 0.771 | 0.393 | 0.760 | 0.662 | 0.872 | 0.350 |
| iTransformer | 0.524 | 0.546 | 0.724 | 0.464 | 0.561 | 0.568 | 0.749 | 0.426 | 0.738 | 0.652 | 0.859 | 0.372 |
| PatchTST | 0.535 | 0.552 | 0.731 | 0.453 | 0.571 | 0.573 | 0.756 | 0.416 | 0.743 | 0.655 | 0.862 | 0.366 |
| TimesNet | 0.550 | 0.562 | 0.742 | 0.438 | 0.586 | 0.583 | 0.766 | 0.401 | 0.754 | 0.659 | 0.869 | 0.356 |
| TSMixer | 0.529 | 0.549 | 0.727 | 0.459 | 0.565 | 0.570 | 0.752 | 0.422 | 0.736 | 0.651 | 0.858 | 0.374 |
| *Foundation models and RGMR* | | | | | | | | | | | | |
| TabPFN | 0.462 | 0.503 | 0.680 | 0.521 | 0.389 | 0.445 | 0.624 | 0.632 | 0.566 | 0.562 | 0.752 | 0.442 |
| TabPFN + RGMR | 0.344 | 0.451 | 0.587 | 0.615 | 0.305 | 0.401 | 0.552 | 0.713 | 0.452 | 0.489 | 0.672 | 0.541 |
| TimeGPT | 0.437 | 0.489 | 0.661 | 0.536 | 0.351 | 0.431 | 0.592 | 0.658 | 0.543 | 0.551 | 0.737 | 0.456 |
| TimeGPT + RGMR | 0.325 | 0.437 | 0.570 | 0.632 | 0.278 | 0.386 | 0.527 | 0.738 | 0.431 | 0.475 | 0.657 | 0.562 |
| TimesFM | 0.391 | 0.465 | 0.625 | 0.564 | 0.318 | 0.412 | 0.564 | 0.687 | 0.524 | 0.536 | 0.724 | 0.469 |
| TimesFM + RGMR | **0.318** | **0.423** | **0.561** | **0.651** | **0.258** | **0.334** | **0.458** | **0.746** | **0.426** | **0.436** | **0.589** | **0.568** |

$\eta^{(k)} \in (0, 2]$ and $\mathbf{w}_i^{(k-1)} \in [0, 1]$ (Theorem 1), and the multi-level behavior exhibits geometric decay to a noise floor as predicted by the corollary.

Table 3: Ablation of RGMR components on one-month-ahead SPEI forecasting (averaged across three South Australian sites).

| Method | TabPFN | | | | TimeGPT | | | | TimesFM | | | |
|---|---|---|---|---|---|---|---|---|---|---|---|---|
| | MSE↓ | MAE↓ | RMSE↓ | R²↑ | MSE↓ | MAE↓ | RMSE↓ | R²↑ | MSE↓ | MAE↓ | RMSE↓ | R²↑ |
| Foundation Model (frozen) | 0.472 | 0.503 | 0.685 | 0.532 | 0.444 | 0.490 | 0.663 | 0.550 | 0.411 | 0.471 | 0.638 | 0.573 |
| + Coarse projection only | 0.436 | 0.482 | 0.662 | 0.570 | 0.410 | 0.467 | 0.641 | 0.590 | 0.387 | 0.448 | 0.610 | 0.604 |
| + Multi-resolution (no refinement) | 0.414 | 0.469 | 0.646 | 0.592 | 0.390 | 0.455 | 0.626 | 0.615 | 0.372 | 0.433 | 0.592 | 0.627 |
| + RGMR (no weights) | 0.380 | 0.459 | 0.620 | 0.609 | 0.360 | 0.442 | 0.602 | 0.625 | 0.341 | 0.405 | 0.548 | 0.646 |
| + RGMR (Full) | **0.367** | **0.447** | **0.604** | **0.623** | **0.345** | **0.433** | **0.585** | **0.644** | **0.334** | **0.398** | **0.536** | **0.655** |

The results reveal several insights about the contribution of individual components. Even using a single resolution (coarse level only) provides noticeable improvements over the base models, confirming our hypothesis that coarse resolution can effectively capture dominant low-frequency patterns in SPEI data. Adding multiple resolutions without residual guidance further enhances performance, demonstrating the value of capturing information at different temporal scales. Incorporating residual feedback (RGMR without weights) yields substantial additional gains, validating the effectiveness of our iterative refinement strategy. The full RGMR method with adaptive residual weights achieves the best performance across all metrics, confirming the importance of weighted residual attention in guiding the refinement process. These findings empirically validate the theoretical foundations of our approach outlined in Section 4, particularly the geometric error contraction demonstrated in Theorem 1.

# 6 CONCLUSION

This paper introduced RGMR, a residual-guided multi-resolution refinement framework that adapts frozen time series foundation models to regional drought forecasting. RGMR iteratively projects predictions across multiple temporal scales and corrects them with predicted residuals, allowing the model to capture long-term climate structure and short-term variability without updating base model parameters. We established theoretical guarantees for geometric error contraction across refinement levels until reaching a noise-determined floor. On South Australian SPEI benchmarks, RGMR improves mean squared error by up to 18.9 percent over strong baselines. The method is training-free for the foundation model and relies only on lightweight Ridge regression for residual prediction, making it computationally efficient and ready for operational drought monitoring.

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

## A  NOTATION AND PROJECTION OPERATORS

### A.1  NOTATION

Table 4: Notation used throughout. *Scalars*: lowercase roman; *vectors*: bold lowercase; *matrices*: bold uppercase; *sets & operators*: calligraphic. All random variables are understood at a fixed forecast origin $t$ unless stated.

| Symbol | Description |
|---|---|
| $N$ | length of the full time series (samples) |
| $W,\ D,\ H$ | input window length, feature dimension, forecast horizon |
| $\{\mathbf{X}_1, \ldots, \mathbf{X}_N\}$ | multivariate climate time series |
| $\mathbf{X}_t \in \mathbb{R}^D$ | feature vector at time $t$ (first entry is SPEI) |
| $\mathbf{X}_{t-W+1:t} \in \mathbb{R}^{W \times D}$ | standardized input window ending at $t$ (train stats only) |
| $\mathbf{y}_{t+1:t+H} \in \mathbb{R}^H$ | target vector for steps $t+1$ to $t+H$ |
| $\hat{\mathbf{y}}^{(k)}_{t+1:t+H}$ | level-$k$ forecast (after refinement level $k$) |
| $f_\theta$ | frozen foundation model (no parameter updates at inference) |
| $\{g^{(k)}_\phi\}^K_{k=1}$ | ridge residual predictors (one per level, trained on training split) |
| $\mathcal{R} = \{r_1 > \cdots > r_K\}$ | resolution (stride) ladder from coarse to fine; $K = |\mathcal{R}|$ |
| $\mathcal{D}_r,\ \mathcal{U}_r$ | down-/up-projection at stride $r$ (block average, repeat) |
| $\mathcal{P}_r = \mathcal{U}_r \circ \mathcal{D}_r$ | projection operator at stride $r$ |
| $C^{(k)}$ | level-$k$ proposal: $C^{(k)} = f_\theta(\mathcal{P}_{r_k}(\mathbf{X}_{t-W+1:t}))$ |
| $\tilde{\mathbf{y}}^{(k)}_{t+1:t+H}$ | *short-window* base forecast used to form training residuals |
| $\tilde{\mathbf{R}}^{(k)}$ | *training* residual at level $k$: $\tilde{\mathbf{R}}^{(k)} = \mathbf{y} - \tilde{\mathbf{y}}^{(k)}$ |
| $\hat{\tilde{\mathbf{R}}}^{(k)}$ | predicted training residual used at inference (from $g^{(k)}_\phi$) |
| $\mathbf{R}^{(k)}$ | *inference* residual at level $k$: $\mathbf{R}^{(k)} = \mathbf{y} - \hat{\mathbf{y}}^{(k)}$ |
| $\mathbf{w}^{(k)} \in [0,1]^H$ | residual-magnitude weights at level $k$; $\mathbf{w}^{(k)} = \mathrm{clip}_{[\varepsilon_{\min}, 1]}\big(\sigma(\gamma(|\hat{\tilde{\mathbf{R}}}^{(k)}| - \delta^{(k)}))\big)$ |
| $\gamma$ | weight sharpness (sigmoid slope), small constant (default $= 3.0$) |
| $\delta^{(k)}$ | level-$k$ residual threshold (validation-selected quantile of $|\hat{\tilde{\mathbf{R}}}^{(k)}|$) |
| $\varepsilon_{\min}$ | weight floor for numerical stability (default $10^{-3}$) |
| $\eta^{(k)},\ \alpha^{(k)}$ | step size and mixing coefficient at level $k$ ($0 < \alpha^{(k)} < 1$, coarse→fine increasing) |
| $\rho^{(k)}$ | contraction modulus: $(1 - \alpha^{(k)}) \max_i |1 - \eta^{(k)} W^{(k-1)}_i|$ |

*Unification note.* We write the validation-percentile threshold uniformly as $\delta^{(k)} \equiv \tau^{(k)}_{\mathrm{dynamic}}$; the logistic uses $\sigma(x) = 1/(1 + e^{-x})$ with fixed sharpness $\gamma = 3.0$; weights are clipped elementwise to $[\varepsilon_{\min}, 1]$ with $\varepsilon_{\min} > 0$ by default.

**Conventions.** $\| \cdot \|_2$ is the Euclidean norm for vectors and the spectral (operator) norm for matrices. $\odot$ is elementwise (Hadamard) product; $\mathrm{Diag}(\mathbf{v})$ forms a diagonal matrix; $\sigma(\cdot)$ is the logistic function.

### A.2  PROJECTION USED IN THE MAIN EXPERIMENTS (SIMPLE AVERAGE & REPEAT)

All main results use the simple projection

$$\mathcal{P}_r = \mathcal{U}_r \circ \mathcal{D}_r,$$

where $\mathcal{D}_r$ averages non-overlapping blocks of length $r$ and $\mathcal{U}_r$ repeats each block average $r$ times. All channels share the same timing to preserve cross-channel coherence. This projection is deterministic, lightweight ($O(W)$ per scale), and is used *throughout the paper and all ablations* unless stated otherwise.

**Lemma 1 (Boundedness of the simple projection)** *Let $D_r$ be block-averaging (row-stochastic) and $U_r$ be repeat-upsampling. Then $\|D_r\|_2 \le 1$, $\|U_r\|_2 \le \sqrt{r}$, and hence $\|\mathcal{P}_r\|_2 = \|U_r D_r\|_2 \le \|U_r\|_2 \|D_r\|_2 \le \sqrt{r}$.*

# B  PROOFS AND TECHNICAL DETAILS

## B.1  RGMR UPDATE AND RESIDUAL RECURSION

We restate the RGMR update in Section 4 (suppressing the origin index $t$ for clarity):

$$\hat{\mathbf{y}}^{(k)} = \alpha^{(k)} C^{(k)} + (1-\alpha^{(k)})\Big(\hat{\mathbf{y}}^{(k-1)} + \eta^{(k)} \mathbf{w}^{(k-1)} \odot \widehat{\mathbf{R}^{(k-1)}}\Big), \qquad k \ge 2, \quad \hat{\mathbf{y}}^{(1)} = C^{(1)}. \quad (8)$$

Here $\widehat{\mathbf{R}^{(k-1)}} = \mathbf{R}^{(k-1)} + \mathbf{e}^{(k-1)}$ with $\mathbf{R}^{(k-1)} = \mathbf{y} - \hat{\mathbf{y}}^{(k-1)}$ and $\mathbf{e}^{(k-1)}$ denoting residual-prediction error (zero mean, bounded second moment). Let $D^{(k-1)} = \mathrm{Diag}(\mathbf{w}^{(k-1)})$. Subtracting equation 8 from $\mathbf{y}$ yields

$$\mathbf{R}^{(k)} = (1-\alpha^{(k)})\big(I - \eta^{(k)} D^{(k-1)}\big)\mathbf{R}^{(k-1)} - (1-\alpha^{(k)})\eta^{(k)} D^{(k-1)} \mathbf{e}^{(k-1)} + \alpha^{(k)} \underbrace{(\mathbf{y} - C^{(k)})}_{\mathbf{u}^{(k)}}. \quad (9)$$

**Lemma 2 (Diagonal operator norm identity)** *If $D$ is diagonal with entries in $[0,1]$ and $\eta > 0$, then $\big\|I - \eta D\big\|_2 = \max_i |1 - \eta D_{ii}|$.*

**Proof 1** *$I - \eta D$ is diagonal; its spectral norm equals the largest absolute diagonal entry.*

## B.2  QUAD PROOF OF THEOREM 1

Define

$$A^{(k)} = (1-\alpha^{(k)})(I - \eta^{(k)} D^{(k-1)}), \quad B^{(k)} = -(1-\alpha^{(k)})\eta^{(k)} D^{(k-1)}, \quad d^{(k)} = \alpha^{(k)} \mathbf{u}^{(k)}.$$

Then equation 9 is $\mathbf{R}^{(k)} = A^{(k)} \mathbf{R}^{(k-1)} + B^{(k)} \mathbf{e}^{(k-1)} + d^{(k)}$. By Lemma 2,

$$\|A^{(k)}\|_2 = (1-\alpha^{(k)})\big\|I - \eta^{(k)} D^{(k-1)}\big\|_2 = (1-\alpha^{(k)}) \max_i |1 - \eta^{(k)} \mathbf{w}_i^{(k-1)}| =: \rho^{(k)}.$$

With $\alpha^{(k)} \in (0,1)$, $\eta^{(k)} \in (0,2]$, and $\mathbf{w}_i^{(k-1)} \in [0,1]$, we have $|1 - \eta^{(k)} \mathbf{w}_i^{(k-1)}| \le 1$, hence $\rho^{(k)} \le 1 - \alpha^{(k)} < 1$.

Next, use the two-parameter Young inequality twice: for any $\delta_1, \delta_2 > 0$ and vectors $u, v, w$,

$$\|u + v + w\|_2^2 \le (1+\delta_1)\|u\|_2^2 + (1+\tfrac{1}{\delta_1})(1+\delta_2)\|v\|_2^2 + (1+\tfrac{1}{\delta_1})(1+\tfrac{1}{\delta_2})\|w\|_2^2.$$

Apply this to $u = A^{(k)} \mathbf{R}^{(k-1)}$, $v = B^{(k)} \mathbf{e}^{(k-1)}$, $w = d^{(k)}$, take expectations, and bound norms:

$$\mathbb{E}\|\mathbf{R}^{(k)}\|_2^2 \le (1+\delta_1)\|A^{(k)}\|_2^2 \, \mathbb{E}\|\mathbf{R}^{(k-1)}\|_2^2 + (1+\tfrac{1}{\delta_1})(1+\delta_2)\|B^{(k)}\|_2^2 \, \mathbb{E}\|\mathbf{e}^{(k-1)}\|_2^2$$
$$+ (1+\tfrac{1}{\delta_1})(1+\tfrac{1}{\delta_2})\|d^{(k)}\|_2^2.$$

**Explicit, level-adaptive constants.** Choose $\boxed{\delta_1 = \alpha^{(k)}, \ \delta_2 = 1}$. Then

$$(1 + \delta_1)\|A^{(k)}\|_2^2 = (1+\alpha^{(k)})(\rho^{(k)})^2 \le (1+\alpha^{(k)})(1-\alpha^{(k)})^2 = 1 - \alpha^{(k)} - \alpha^{(k)2} < 1,$$

so the leading factor remains a strict contraction *without* any redefinition of $\rho^{(k)}$. Moreover,

$$\|B^{(k)}\|_2 \le (1-\alpha^{(k)})\eta^{(k)}\|D^{(k-1)}\|_2 \le (1-\alpha^{(k)})\eta^{(k)}, \qquad \|d^{(k)}\|_2 = \alpha^{(k)}\|\mathbf{u}^{(k)}\|_2.$$

Hence we obtain

$$\mathbb{E}\|\mathbf{R}^{(k)}\|_2^2 \le \underbrace{\left[(1+\alpha^{(k)})(\rho^{(k)})^2\right]}_{<1} \mathbb{E}\|\mathbf{R}^{(k-1)}\|_2^2 + B_k, \tag{10}$$

with the explicit bound

$$B_k \le 2\left(1+\tfrac{1}{\alpha^{(k)}}\right)(1-\alpha^{(k)})^2(\eta^{(k)})^2\sigma_e^2 + 2\left(1+\tfrac{1}{\alpha^{(k)}}\right)(\alpha^{(k)})^2\mathbb{E}\|\mathbf{u}^{(k)}\|_2^2, \tag{11}$$

where $\sigma_e^2 \ge \mathbb{E}\|\mathbf{e}^{(k-1)}\|_2^2$. This proves Theorem 1 with fully spelled-out constants.

**Optional simplified constants (4 and 4).** If one *strengthens* backtracking to enforce $\rho^{(k)} \le 1/\sqrt{2}$ (instead of merely $\rho^{(k)} < 1$), then setting $\delta_1 = \delta_2 = 1$ gives

$$\mathbb{E}\|\mathbf{R}^{(k)}\|_2^2 \le \underbrace{2(\rho^{(k)})^2}_{\le 1} \mathbb{E}\|\mathbf{R}^{(k-1)}\|_2^2 + \underbrace{4(1-\alpha^{(k)})^2(\eta^{(k)})^2\sigma_e^2}_{\text{noise}} + \underbrace{4(\alpha^{(k)})^2\mathbb{E}\|\mathbf{u}^{(k)}\|_2^2}_{\text{proposal}},$$

i.e.,

$$B_k \le 4(1-\alpha^{(k)})^2(\eta^{(k)})^2\sigma_e^2 + 4(\alpha^{(k)})^2\mathbb{E}\|\mathbf{u}^{(k)}\|_2^2.$$

This "clean-4" form is sometimes convenient in practice, at the cost of a slightly stronger backtracking criterion.

**Modeling note (Ridge $\Rightarrow$ bounded noise).** In code, $\mathbf{e}^{(k-1)}$ is precisely the prediction error of the (deterministic) Ridge regressor $g_\phi$ on level $k-1$. Treating it as zero-mean with bounded second moment is standard when the residual regressor has limited capacity relative to the true residual process and is trained on a stationary slice; the variance proxy $\sigma_e^2$ is estimated on validation.

### B.3 PROOF OF COROLLARY 1

**Proof 2** *We proceed in three steps.*

**Step 1: Invoke the per-level bound at level $K$.** *By the update rule and the non-expansive step-size condition in §4.5, the level-$K$ prediction satisfies the per-level $\ell_\infty$ bound*

$$\|\mathbf{y}^\star - \hat{\mathbf{y}}^{(K)}\|_\infty \le (1-\alpha^{(K)}) \underbrace{\max_{j\in[H]} \left|1 - \eta^{(K)} w_{t+1:t+H,\,j}^{(K-1)}\right|}_{\text{non-expansive factor}} \|\mathbf{y}^\star - \hat{\mathbf{y}}^{(K-1)}\|_\infty$$

$$+ \alpha^{(K)} B_K + (1-\alpha^{(K)})\eta^{(K)}\|\mathbf{w}^{(K-1)}\|_\infty E_{K-1}. \tag{12}$$

*This is exactly the $k{=}K$ instance of the general per-level inequality in §4.5, obtained by: (i) subtracting $\mathbf{y}^\star$ from the level-$K$ update, (ii) decomposing the residual prediction as $\widehat{\mathbf{R}}^{(K-1)} = \tilde{\mathbf{R}}^{(K-1)} + \varepsilon^{(K-1)}$ with $\|\varepsilon^{(K-1)}\|_\infty \le E_{K-1}$, (iii) applying the operator norm identity $\|I - \eta^{(K)}\mathrm{Diag}(\mathbf{w}^{(K-1)})\|_{\infty\to\infty} = \max_j |1 - \eta^{(K)}w_j^{(K-1)}|$, and (iv) using $\|\mathbf{a}\odot\mathbf{b}\|_\infty \le \|\mathbf{a}\|_\infty\|\mathbf{b}\|_\infty$ together with the backbone error bound $\|\mathbf{y}^\star - \mathbf{C}^{(K)}\|_\infty \le B_K$.*

**Step 2: Upper bound for the direct finest-level baseline.** *By the definition of $B_K$ (the $\ell_\infty$ upper bound on the level-$K$ backbone error), the direct one-pass baseline obeys*

$$\|\mathbf{y}^\star - \hat{\mathbf{y}}_{\mathrm{dir}}\|_\infty = \|\mathbf{y}^\star - \mathbf{C}^{(K)}\|_\infty \le B_K. \tag{13}$$

**Step 3: Sufficient condition implying RGMR $\le$ direct.** *Starting from equation 12, drop the middle term $\alpha^{(K)} B_K$ to the right-hand side of the desired comparison with equation 13. It is therefore sufficient that the remaining two terms are together no larger than $B_K$, i.e.,*

$$\max_{j\in[H]} \left|1 - \eta^{(K)} w_{t+1:t+H,\,j}^{(K-1)}\right| \|\mathbf{y}^\star - \hat{\mathbf{y}}^{(K-1)}\|_\infty + \eta^{(K)}\|\mathbf{w}^{(K-1)}\|_\infty E_{K-1} \le B_K. \tag{14}$$

*Under equation 14, inequality equation 12 yields*

$$\|\mathbf{y}^\star - \hat{\mathbf{y}}^{(K)}\|_\infty \le \alpha^{(K)} B_K + \left(B_K - \alpha^{(K)} B_K\right) = B_K,$$

*and combining with equation 13 gives $\|\mathbf{y}^\star - \hat{\mathbf{y}}^{(K)}\|_\infty \le \|\mathbf{y}^\star - \hat{\mathbf{y}}_{\mathrm{dir}}\|_\infty$, which proves the claim.*

## B.4 Complexity and Boundedness

For $\mathcal{P}_r = \mathcal{U}_r \circ \mathcal{D}_r$ (block average + repeat), each scale costs $O(W)$ per window; Ridge inference is negligible vs. one $f_\theta$ call. The total cost per origin is $O(|\mathcal{R}| \cdot C_\theta + |\mathcal{R}| \cdot W)$, where $C_\theta$ is the cost of one call to $f_\theta$. Boundedness of $\mathcal{P}_r$ is immediate since block averaging is non-expansive in $\ell_2$ and repeating has finite gain (at most $\sqrt{r}$), thus uniformly bounded over the fixed ladder $\mathcal{R} = \{12, 6, 3, 2, 1\}$.

## C  Datasets, Regions, Covariates, and Splits

**Regions and time span.**  We evaluate regional SPEI forecasting at three representative sites in South Australia. Table 9 lists coordinates, nominal climate zones, coverage, and sample counts. Coordinates refer to the center of the grid cell used to extract both covariates and the target series. Monthly cadence from **1982/01 to 2018/12** yields **444** samples per site.

Table 5: Regions, climate zones, coverage, and sample counts at monthly resolution.

| Region ID | Coordinates (lat, lon) | Climate zone | Years covered | Total samples |
|---|---|---|---|---|
| SA-1 | $(-26.125, \ 129.125)$ | Arid | 1982 to 2018 | 444 |
| SA-2 | $(-29.125, \ 134.875)$ | Arid and seasonal | 1982 to 2018 | 444 |
| SA-3 | $(-35.625, \ 138.875)$ | Temperate | 1982 to 2018 | 444 |

**Target and data sources.**  The forecasting target is **SPEI-30**, the Standardized Precipitation Evapotranspiration Index aggregated over a 30 month window at monthly resolution. Positive values indicate wetter than normal conditions and negative values indicate drier than normal conditions. Target series are taken from the *SPEI Global Drought Monitor* at $0.5° \times 0.5°$ monthly resolution, sampled at the grid cell centers in Table 9. Meteorological covariates are taken from *NCEP NCAR Reanalysis 1* at $2.5° \times 2.5°$ monthly resolution and are sampled or bilinearly interpolated to the site cells as needed. Large scale climate indices are obtained from public operational releases including Niño 1 plus 2, Niño 3.4, Niño 4, the Dipole Mode Index for the Indian Ocean Dipole, and the Southern Annular Mode. All sources are publicly available. The SPEI product is accessible at `https://doi.org/10.5281/zenodo.8060268`. NCEP NCAR Reanalysis 1 documentation and downloads are available from NOAA at `https://www.psl.noaa.gov/data/gridded/data.ncep.reanalysis.html`. Public climate indices are available from operational centers such as NOAA CPC.

**Covariate composition.**  Each input vector $\mathbf{X}_t \in \mathbb{R}^D$ contains the most recent SPEI-30 value, lagged SPEI-30 within the history window, and local meteorological covariates drawn from NCEP NCAR Reanalysis 1 including precipitation, mean temperature, maximum temperature, minimum temperature, potential evapotranspiration, and vapor pressure deficit. Monthly large scale indices are included as scalar drivers and are replicated across the history window when constructing inputs so that all channels share the same timestamps. All variables are aligned to calendar months.

**Acquisition and spatial sampling.**  SPEI-30 is downloaded at $0.5°$ spatial resolution and monthly cadence, then extracted at the study cell centers. NCEP NCAR Reanalysis 1 fields are downloaded at $2.5°$ spatial resolution and monthly cadence for the South Australia bounding box. When a variable is available on a coarser grid, values are taken from the nearest cell or from bilinear interpolation to the site coordinates. Climate indices are downloaded as monthly series and aligned to the target timeline.

**Rolling origin evaluation and splits.**  We use a rolling origin evaluation with non overlapping test origins. Training, validation, and test spans are chronological, and the step between consecutive test origins equals the forecast horizon, that is $s = H$ for $H \in \{1, 2, 3\}$. Per channel standardization is fit on the training span only and applied unchanged to validation and test. No information from validation or test targets is used in preprocessing or covariate construction. The resolution ladder and all hyperparameters are held fixed across regions and splits with no retuning.

**Preprocessing and quality control.** Within channel missing values are linearly interpolated. Terminal gaps of at most two months are back filled or forward filled. After standardization, values beyond five median absolute deviations are clipped to the nearest bound. Calendar alignment ensures that the last index in each input window coincides with the forecast origin month. All transformations are fit on the training span only and then reused without modification on validation and test.

**Evaluation protocol.** For each region and each horizon we compute MSE, MAE, RMSE, and $R^2$ per test origin. We then aggregate per region by the median and the interquartile range across origins. In addition to absolute scores we report skill relative to a climatology baseline defined as the rolling mean computed on the training span only. The foundation baseline, the resolution ladder, and the backtracking criteria are identical to those used in the main text.

**Data availability and licensing.** The SPEI Global Drought Monitor is distributed under a Creative Commons Attribution license at $0.5° \times 0.5°$ monthly resolution. NCEP NCAR Reanalysis 1 meteorological fields at $2.5° \times 2.5°$ monthly resolution are public domain through NOAA. Public climate indices from operational centers are also publicly accessible. All series used in this study are obtained from public endpoints and do not require bespoke access.

**Computing environment and runtime.** Experiments are run on a single workstation equipped with an NVIDIA GeForce RTX 3090 graphics processor with 24 GB of memory, an Intel 12700K central processor, 64 GB of DDR4 3200 memory, and a 2 TB NVMe solid state drive. The operating system is Ubuntu 20.04.3 LTS. The software stack uses Python 3.10 with PyTorch 2.0.1 with CUDA 11.8, NumPy 1.24.3, Pandas 2.0.2, and Scikit learn 1.2.2. Random seeds are fixed for NumPy and PyTorch so that runs are deterministic where supported. The anonymized repository includes configuration files and single command scripts that reproduce all scores and figures reported in the paper without retuning.

# D IMPLEMENTATION DETAILS AND HYPERPARAMETERS

**Conventions (global).** All normalization (mean/variance, scaling, and any per-feature statistics) is computed on the *training split only* and reused for validation/test to avoid leakage. Validation is used only to select *thresholds or quantiles* (e.g., $\tau_{\text{dynamic}}^{(k)}$) and other *hyperparameters* as specified below; it never contributes statistics that rescale inputs or targets.

## D.1 PROJECTION OPERATOR, WINDOWING, AND BOUNDARY HANDLING (MAIN)

Unless marked otherwise, all experiments use $\mathcal{P}_r = \mathcal{U}_r \circ \mathcal{D}_r$, where $\mathcal{D}_r$ averages non-overlapping blocks of stride $r$ and $\mathcal{U}_r$ repeats each block average $r$ times. All channels share the same timing to preserve cross-channel coherence.

*Context windowing.* At inference, the long context length is set to $W = \lfloor 0.7\,T \rfloor$ (where $T$ is the available series length at prediction time) so that the base TSFM receives a stable long-range context; the short window for learning residuals follows the per-level stride $r_k$ (Sec.4) and is independent of $W$.

## D.2 RESIDUAL PREDICTOR AND FEATURES

The residual predictor $g_\phi$ is a ridge regressor trained on *training-split* residuals only (with inputs normalized by training statistics). Per horizon, features include: (i) $p$ recent residual lags ($p \in \{3, 6\}$), (ii) rolling mean/std over a short window, (iii) a linear trend coefficient over $q$ recent steps ($q \in \{6, 12\}$), (iv) the current level index (categorical embedding or one-hot). The ridge penalty $\lambda_{\text{ridge}}$ is chosen by time-ordered CV on the validation split from a log-grid $10^{[-4,\ldots,2]}$. *Implementation note.* $\mathbf{z}_t$ concatenates recent targets, simple rolling statistics/trend, a short residual history, and the level index; typical ranges are small (e.g., $p \in \{3, 6\}$, $q \in \{6, 12\}$) and were selected on the validation split.

## D.3 WEIGHTING, BACKTRACKING, AND EARLY STOPPING

*Backtracking and stopping.* At level $k$, we pick $\eta^{(k)} \in (0, 2]$ by a simple backtracking rule to satisfy the per-level contraction $\rho^{(k)} = (1 - \alpha^{(k)}) \max_i |1 - \eta^{(k)} \mathbf{w}_{t,i}^{(k-1)}| < 1$ (defaults: $\eta_{\text{init}} = 1.0$, shrink $\beta = 0.5$, floor $\eta_{\min} = 10^{-3}$). Early stopping triggers when the per-level update norm stabilizes relative to the target scale or the running decrease falls below a small fraction of the Appendix B bound.

**Residual-adaptive weighting (matches main text).** At level $k$, elementwise weights are

$$\mathbf{w}_i^{(k)} = \sigma\Big(3\big(|\widehat{R}_i^{(k)}| - \tau_{\text{dynamic}}^{(k)}\big)\Big), \qquad \sigma(x) = \frac{1}{1+e^{-x}}, \quad \mathbf{w}_i^{(k)} \in [\varepsilon_{\min}, 1],$$

where the logistic sharpness constant "3" is fixed (as in Sec. 4.4), $\tau_{\text{dynamic}}^{(k)}$ is a *validation-selected* quantile of the absolute residual magnitude at level $k$ (quantile grid $\{0.60, 0.70, 0.75, 0.80, 0.85, 0.90\}$ unless stated), and $\varepsilon_{\min} > 0$ prevents zero weights.

**Backtracking step size.** Initialize $\eta^{(k)} \leftarrow \eta_{\text{init}} \in (0, 2]$ and repeatedly halve until the contraction modulus

$$\rho^{(k)} = (1 - \alpha^{(k)}) \max_i |1 - \eta^{(k)} W_i^{(k-1)}| < 1,$$

is satisfied, respecting a floor $\eta_{\min}$ and a cap $J_{\max} = \lceil \log_2(\eta_{\text{init}}/\eta_{\min}) \rceil$ on halvings. This enforces the condition in Theorem 1 with $\eta^{(k)} \in (0, 2]$ and $W^{(k-1)} \in [\varepsilon_{\min}, 1]$.

**Early stopping across levels.** We monitor the per-level validation RMSE decrease and stop refinement when the marginal gain drops below a proxy for $\sqrt{B_k}$ computed from a noise estimate $\widehat{\sigma}_e^2$ (times a data-dependent factor calibrated on validation). This implements the geometric saturation criterion in the corollary to Theorem 1.

## D.4 DEFAULT HYPERPARAMETERS AND RANGES

Table 6: Default hyperparameters and ranges (used unless stated otherwise).

| Parameter | Default | Range/Selection |
|---|---|---|
| Resolution ladder $\mathcal{R}$ | $\{12, 6, 3, 2, 1\}$ | fixed |
| **Logistic sharpness (fixed)** | **3.0** | **fixed constant (matches main text)** |
| **Weight threshold $\tau_{\text{dynamic}}^{(k)}$** | **val-quantile** | **grid $\{0.60, \ldots, 0.90\}$ (per level $k$)** |
| Weight floor $\varepsilon_{\min}$ | $10^{-3}$ | $[0, 10^{-3}]$ |
| Initial step $\eta_{\text{init}}$ | 1.0 | $(0, 2]$ |
| Min step $\eta_{\min}$ | 0.0625 | $\{0.03125, 0.0625, 0.125\}$ |
| Mixing $\alpha^{(k)}$ | formula below | implied in $[0.3, 0.8]$ |
| Residual lags $p$ | 6 | $\{3, 6\}$ |
| Trend window $q$ | 12 | $\{6, 12\}$ |
| Ridge penalty $\lambda_{\text{ridge}}$ | CV-selected | $10^{[-4,\ldots,2]}$ (time-ordered CV) |

## D.5 MIXING COEFFICIENT (MONOTONE COARSE→FINE)

The mixing coefficient increases with finer resolutions and satisfies $0 < \alpha^{(k)} < 1$:

$$\alpha^{(k)} = 0.3 + 0.5\left(1 - \frac{r_k}{\max(\mathcal{R})}\right),$$

where $r_k$ is the current stride and $\max(\mathcal{R}) = 12$ for $\mathcal{R} = \{12, 6, 3, 2, 1\}$, yielding $\alpha^{(k)} \in [0.3, 0.8]$. This choice aligns with the theory by placing more trust on finer-scale proposals while retaining contraction.

## D.6 ADAPTIVE LEARNING RATE

The practical update for $\eta^{(k)}$ combines a convergence-driven term and a residual-magnitude-driven term while respecting the backtracking envelope:

$$\eta^{(k)} \;=\; \alpha_{\mathrm{lr}}\,\eta^{(k)}_{\mathrm{conv}} \;+\; (1 - \alpha_{\mathrm{lr}})\,\eta^{(k)}_{\mathrm{res}}, \qquad \alpha_{\mathrm{lr}} = 0.7,$$

where both components are *monotone* in observed improvement and residual magnitude, respectively, and the final $\eta^{(k)}$ is clipped to $(0, 2]$ and then subject to the backtracking test $\rho^{(k)} < 1$ above. Exact rules match the code release.

## D.7 REMARKS ON HYPERPARAMETER ECONOMY

RGMR uses a small set of hyperparameters with default values (Tab. 6). When selection is required (e.g., $\tau^{(k)}_{\mathrm{dynamic}}$ quantiles, $\lambda_{\mathrm{ridge}}$), we employ a *single* time-ordered validation pass without test-time peeking. This keeps deployment overhead low while preserving the "plug-and-play" nature of the wrapper.

# E WHY SOUTH AUSTRALIA AND SOCIETAL IMPACT

**Rationale for choosing South Australia (SA).**   South Australia offers a stringent and policy relevant testbed for inference time adaptation because it concentrates diverse hydroclimate regimes within a compact domain. Coastal areas experience Mediterranean like seasonality and strong oceanic modulation, while inland basins transition rapidly to semiarid and arid conditions with intermittent rainfall and long dry spells. This sharp coastal to interior gradient produces multiscale temporal variability, where seasonal cycles interact with subseasonal bursts and multiyear swings driven by large scale climate modes such as ENSO, IOD, and SAM. These conditions routinely expose where single resolution forecasters underreact or overreact. Monthly drought relevant targets such as SPEI are characteristically zero inflated and heavy tailed, with long quiescent runs punctuated by clustered extremes. This is precisely the setting where our residual guided, multiresolution refinement can correct coarse, temporally smeared signals without retraining the underlying foundation model. From a practical standpoint, SA is covered consistently by public gridded reanalysis and forecast products at monthly cadence, so we can evaluate the same frozen base model and the same resolution ladder across many locations under a standard rolling origin protocol, avoiding bespoke data access or regional retuning. In short, SA is not a cherry picked easy case. It compresses coastal influence, interior aridity, and teleconnection exposure into one reproducible sandbox, which makes it an informative proxy for a broad class of water limited regions worldwide.

**Societal impact.**   Sharper regional drought and hydroclimate outlooks translate into concrete decisions across sectors. For water security, earlier and more reliable dry spell signals support allocation planning, reservoir operations, and demand management, which reduces the cost of emergency curtailments. In agriculture and viticulture, month ahead guidance for moisture deficits and heat stress improves planting and harvest scheduling, irrigation timing, and input procurement, helping buffer income volatility in adverse seasons. Risk management agencies gain time to preposition assets for bushfire seasons and for heat health responses, with a focus on outreach to the most vulnerable communities when compounded hazards such as dryness plus heat are likely. Power systems with high renewable penetration benefit from subseasonal guidance to schedule maintenance and hedging, since both demand and variable generation respond to persistent hot and dry spells. Insurers and local governments can stress test budgets and infrastructure maintenance against sequences of below median months rather than isolated events. Many remote and regional communities in arid zones are disproportionately exposed to climate variability. Improving foresight at the horizons where operational choices are actually made enables targeted and equitable adaptation measures instead of reactive crisis management.

**Responsible use and limitations.**   RGMR operates strictly at inference time on frozen foundation models. It cannot remove upstream data biases, coverage gaps, or structural errors in the base forecaster, and it should not be used as the sole basis for high stakes decisions. Best practice is

to treat RGMR as one member in a multimodel and multievidence framework that includes expert judgment, observational context, and sector specific constraints. Reported improvements reflect the fixed resolution ladder and backtracking criteria specified in the main text. Changing those design choices or the evaluation geography may alter performance and uncertainty. To support scrutiny, extension, and safe reuse, we release anonymized code with exact configurations and single command scripts mirroring the paper's rolling origin evaluation, so stakeholders can replicate results and reassess them in their own regional contexts before operational uptake.

## F   MORE EXPERIMENTAL PLOTS

**Setup.**   We report additional one–month–ahead results at Location $(-26.125, 129.125)$ using the same leakage–safe protocol as the main paper (identical train/val/test splits, per–channel standardization fit on train, and $H=1$ unless stated). Figure 5 shows predicted vs. observed SPEI prediction plots for twelve baselines.

**Takeaways.**   (1) Foundation-scale models (e.g., TimesFM) tend to yield the tightest clouds, particularly near mild-to-moderate drought values. (2) Frequency/linear families (DLinear, FiLM) often reduce variance but can under-shoot extremes. (3) Patch- and pyramid-style Transformers (PatchTST, Pyraformer) better capture mesoscale variability yet may show scatter at the tails. (4) Nonstationarity-aware designs narrow bias under regime shifts but remain sensitive to rare events.

**Baselines at a glance.**

- **TimesFM**   A decoder only time series foundation model pretrained on large multi domain corpora. It provides strong zero shot and few shot generalization for point forecasting with long context lengths, and it can be used as a frozen backbone with inference time adaptation. Code: google-research/timesfm.
- **Transformer (vanilla)**   A capacity reference built from standard multi head attention, positional encoding, and feed forward blocks. Complexity scales quadratically with sequence length and it serves as a clean yardstick across datasets. Code: thuml/Time-Series-Library (model `Transformer.py` and unified training scripts).
- **Autoformer**   A decomposition oriented Transformer that models trend and seasonality explicitly and replaces dot product attention with series wise autocorrelation to stabilize long horizon prediction. Code: thuml/Autoformer.
- **Crossformer**   A cross dimension attention architecture for multivariate series. It introduces hierarchical embeddings and patching to capture inter variable relations together with temporal dependencies at reduced cost. Code: Thinklab-SJTU/Crossformer.
- **DLinear**   A minimal linear baseline that first separates trend and seasonal components by moving average then applies two one layer linear heads and sums the results. Despite its simplicity it is competitive on long horizon settings. Code: cure-lab/LTSF-Linear.
- **PatchTST**   A patch based Transformer that segments a long series into temporal patches as tokens and adopts channel independent blocks so that each univariate channel shares weights. This design improves stability and efficiency for long horizons. Code: PatchTST/PatchTST.
- **TimeMixer**   An MLP style time mixing model with multi scale mixers that fuse short and long temporal patterns. It targets strong accuracy with training and inference efficiency and serves as a non attention deep baseline. Code: kwuking/TimeMixer.
- **iTransformer**   An inverted formulation that treats time steps as tokens and embeds along the variable axis. This swap emphasizes temporal relations across long histories while keeping standard Transformer blocks. Code: thuml/iTransformer.
- **Pyraformer**   A pyramidal and dilated attention design that captures long range context with sparse patterns and reduced attention cost. It is suited to very long contexts under limited compute. Code: ant-research/Pyraformer.
- **FiLM**   The frequency improved Legendre memory model that uses Legendre state space projection with frequency domain enhancement and low rank parameterization. It can act as a standalone predictor or a plug in representation module. Code: DAMO-DI-ML/NeurIPS2022-FiLM.

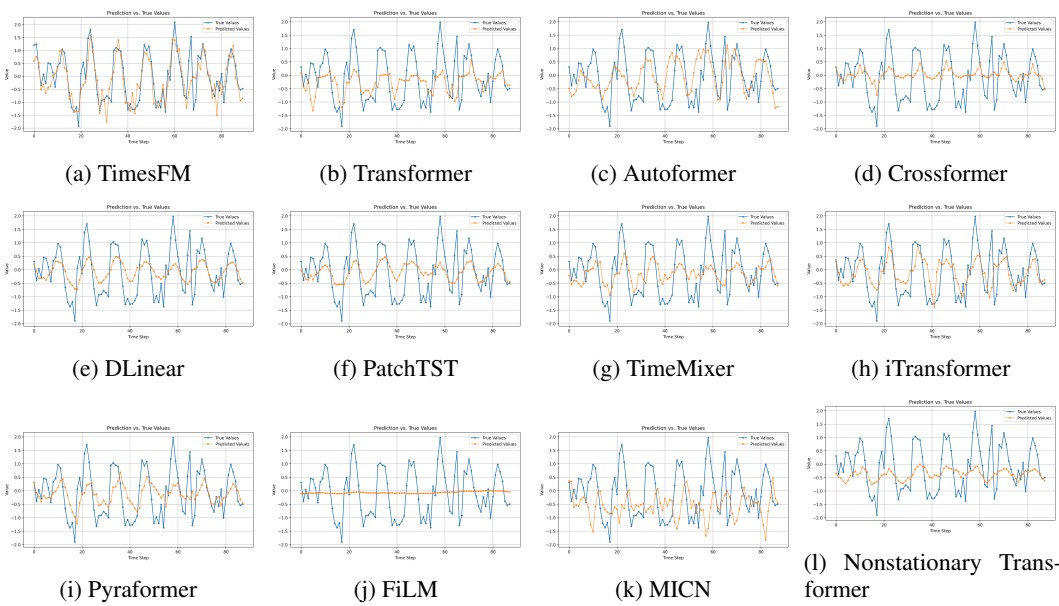

|  |  |  |  |
|---|---|---|---|
| (a) TimesFM | (b) Transformer | (c) Autoformer | (d) Crossformer |
| (e) DLinear | (f) PatchTST | (g) TimeMixer | (h) iTransformer |
| (i) Pyraformer | (j) FiLM | (k) MICN | (l) Nonstationary Transformer |

Figure 5: One–month–ahead SPEI at Location $(-26.125, 129.125)$ (appendix view). Each panel overlays *ground truth* (blue) and *model prediction* (orange) under the same leakage–safe protocol as the main paper (standardization fit on train; identical splits; $H=1$). TimesFM (foundation-scale) shows the tightest phase alignment on mild–moderate variability; linear/frequency families (e.g., DLinear, FiLM) reduce variance but may under-shoot extremes; patch/pyramid Transformers (PatchTST, Pyraformer) capture mesoscale fluctuations with occasional tail scatter; nonstationarity-aware variants alleviate regime-shift bias. No re-tuning is performed for this appendix figure.

- **MICN**  A multi scale model that decomposes series and applies inception style convolutions to capture local details together with global context. It is a strong convolutional style baseline for long term forecasting. Code: wanghq21/MICN.

- **Nonstationary Transformer**  A framework that introduces series stationarization and de stationarized attention to mitigate distribution shifts through time. It can be combined with several attention backbones including the vanilla Transformer. Code: thuml/Nonstationary_Transformers.

**Unified scripts.**  For reviewers who prefer a single entry point, the THUML Time-Series-Library provides consistent data loaders, training scripts, and reference configurations that cover most baselines listed above.

## G  EXTENDED EXPERIMENTS AND CLARIFICATIONS

This appendix provides all extended experiments including: (i) generalization across horizons, variables, regions, and TSFMs; (ii) global verification of contraction; (iii) latency and computational efficiency; (iv) ladder sensitivity and projection operator ablations; and (v) clarification of methodological scope. These results complement the main text and further demonstrate the effect of RGMR.

### G.1  GENERALIZATION ACROSS HORIZONS, VARIABLES, AND REGIONS

**Longer-horizon forecasting.**  Table 7 evaluates RGMR for $H = \{1, 3, 6\}$ months. RGMR consistently improves TimesFM on all horizons.

**Additional variable: Temperature.**  RGMR also improves temperature forecasting (Table 8), confirming that its refinement is not SPEI-specific.

Table 7: Forecasting performance at different horizons.

| Horizon | Model | MSE↓ | MAE↓ | $R^2$↑ |
|---|---|---|---|---|
| 1 | TimesFM | 0.391 | 0.465 | 0.625 |
| | RGMR | **0.318** | **0.423** | **0.651** |
| 3 | TimesFM | 0.438 | 0.502 | 0.575 |
| | RGMR | **0.345** | **0.433** | **0.627** |
| 6 | TimesFM | 0.496 | 0.539 | 0.535 |
| | RGMR | **0.376** | **0.465** | **0.606** |

Table 8: Temperature forecasting results.

| Model | MSE↓ | MAE↓ | $R^2$↑ |
|---|---|---|---|
| TimesFM | 3.121 | 1.534 | 0.933 |
| RGMR | **2.653** | **1.304** | **0.943** |

**Cross-region evaluation.** We then test the generalizations of RGMR by extending to 3 more regions (Table 9).

Table 9: Cross-region generalization.

| Region | Model | MSE↓ | MAE↓ | $R^2$↑ |
|---|---|---|---|---|
| US West Coast | TimesFM | 0.155 | 0.313 | 0.854 |
| | RGMR | **0.0809** | **0.222** | **0.924** |
| North Africa | TimesFM | 0.101 | 0.261 | 0.892 |
| | RGMR | **0.0649** | **0.201** | **0.933** |
| Indonesia | TimesFM | 0.260 | 0.407 | 0.485 |
| | RGMR | **0.204** | **0.348** | **0.598** |

**Additional TSFMs.** To further prove the effectiveness of RGMR, we evaluate performance on two more frozen TSFMs.

Table 10: RGMR improves Chronos and Lag-Llama.

| Model | MSE↓ | MAE↓ | $R^2$↑ |
|---|---|---|---|
| Chronos | 0.401 | 0.475 | 0.740 |
| Chronos + RGMR | **0.343** | **0.439** | **0.780** |
| Lag-Llama | 0.450 | 0.503 | 0.700 |
| Lag-Llama + RGMR | **0.385** | **0.465** | **0.750** |

## G.2 GLOBAL VERIFICATION OF CONTRACTION

We compute the improvement ratio:

$$\frac{1}{N} \sum_{i=1}^{N} \mathbf{1}\{\text{Error}_i^{(k)} < \text{Error}_i^{(k-1)}\}$$

for each refinement layer. Table 11 shows that 40–80% of samples improve per step, providing global empirical support for the theoretical contraction property.

Table 11: Global contraction verification.

| Layer | Scale | Improvement Ratio | Improved / Total |
|-------|-------|-------------------|------------------|
| 2 | 6 | 39.71% | 27/68 |
| 3 | 3 | 79.41% | 54/68 |
| 4 | 2 | 66.18% | 45/68 |
| 5 | 1 | 48.53% | 33/68 |

### G.3 LATENCY AND COMPUTATIONAL EFFICIENCY

Latency measured with `torch.cuda.Event` over 100 runs on RTX 3090 is shown in Table 12. Despite five forward passes, RGMR adds only ∼60ms, negligible for monthly drought forecasting.

Table 12: GPU latency.

| Model | Mean (ms) | Std (ms) |
|-------|-----------|----------|
| TimesFM | 14.52 | 2.18 |
| RGMR | 74.89 | 5.00 |

### G.4 SENSITIVITY TO RESOLUTION LADDERS

Table 13 compares five resolution ladders, including a PSD-derived hierarchy. RGMR is robust across all variants, with only mild variation in accuracy.

Table 13: Ladder sensitivity results.

| Ladder | Scales | MSE↓ | MAE↓ | $R^2$↑ |
|--------|--------|------|------|--------|
| L1 | [12,6,3,2,1] | **0.318** | **0.423** | **0.651** |
| L2 | [16,8,4,2,1] | 0.324 | 0.439 | 0.644 |
| L3 | [12,4,2,1] | 0.322 | 0.438 | 0.645 |
| L4 | [12,6,3,1] | 0.330 | 0.443 | 0.637 |
| L5 | [13,6,4,3,1] | 0.330 | 0.443 | 0.636 |

### G.5 PROJECTION OPERATOR ABLATION

We evaluated three projection operators used for downscaling to further prove the choice: block averaging, wavelet decomposition, and STL decomposition. As shown in Table 14, simple block averaging provides the most stable and accurate refinement, whereas more complex decompositions tend to introduce unnecessary distortions at coarser scales.

Table 14: Projection operator comparison.

| Projection | MSE↓ | MAE↓ | $R^2$↑ |
|-----------|------|------|--------|
| Block Averaging | **0.318** | **0.423** | **0.651** |
| Wavelet | 0.366 | 0.467 | 0.597 |
| STL | 0.387 | 0.473 | 0.574 |

### G.6 SCOPE CLARIFICATION

RGMR works as a refinement mechanism that kicks in during inference with frozen time-series foundation models. Rather than touching model parameters, it operates purely on predictions and works with any TSFM that has a standard forward interface, whether univariate or multivariate. This

makes the refinement architecture-agnostic and ready to use without retraining or parameter tuning for specific tasks.

Let me clarify where RGMR fits in the broader climate forecasting landscape. We evaluate on SPEI forecasting rather than full climate fields for a practical reason: SPEI is a derived index that aggregates multiple meteorological variables through nonlinear transformations. Most operational systems model its temporal behavior directly rather than reconstructing it from raw atmospheric data. Full climate simulators like SEAS5 or NMME need high-dimensional atmospheric fields including temperature, humidity, pressure, and winds that we don't have access to here. They're built for different purposes entirely, so comparing them wouldn't make sense.

It's also worth distinguishing RGMR from test-time adaptation approaches. Methods like PETSA and DynaTTA update parameters online using specialized losses and auxiliary modules. RGMR takes a different route by refining outputs without touching parameters at all. These approaches actually complement each other: TTA modifies the model itself while RGMR modifies the predictions. You could potentially stack RGMR on top of a TTA-adapted model, though we haven't explored that here.

Our experiments show that RGMR delivers consistent gains across different climates, forecast horizons, variables, backbone models, resolution levels, and projection operators. The overhead is minimal, and the results confirm the contraction property our theory predicted.

## H  USE OF LARGE LANGUAGE MODELS

We used a large language model (LLM) solely as a writing assistant to improve grammar, fluency, and consistency of the manuscript text. The LLM did not generate scientific content, equations, algorithms, experimental designs, figures, tables, or results. All ideas, methods, analyses, code, and conclusions are by the authors; all citations and numbers were manually verified. No non-public data were provided to the tool; only de-identified manuscript text was used for editing and sanity checks of mathematical presentation.

## I  ETHICS STATEMENT

This work adheres to the ICLR Code of Ethics. We study an inference-time refinement framework (RGMR) for climate time-series forecasting that *keeps the backbone foundation model $f_\theta$ unchanged* and trains only lightweight ridge regressors to predict residual corrections. Our experiments use publicly available, non-personal environmental datasets; no human subjects or animal experiments were involved, and no personally identifiable or sensitive information was used. We comply with dataset/model licenses by preserving attributions and, where redistribution is restricted, providing scripts to fetch official releases. RGMR is intended to assist scientific analysis rather than serve as the sole basis for high-stakes decisions; domain-expert review and calibration checks are recommended. The environmental impact is modest because no fine-tuning of large backbones is performed.

## J  REPRODUCIBILITY STATEMENT

We aim for full reproducibility. All code, configuration files, random seeds, and data-acquisition scripts (with checksums) will be publicly available; links to the repository and datasets are already provided in the paper. The backbone $f_\theta$ is loaded with unmodified weights; per-resolution ridge regressors use hyperparameters chosen by chronological validation and remain fixed at inference.

