# OpenReview forum: "RESIDUAL-GUIDED MULTI-RESOLUTION REFINEMENT OF FOUNDATION MODELS - A CASE STUDY IN CLIMATE FORECASTING"
_ICLR.cc/2026/Conference — ICLR 2026 Conference Desk Rejected Submission_

### Official Review · Reviewer_RhUR · 2025-10-20

**Soundness:** 2
**Presentation:** 2
**Contribution:** 2
**Rating:** 4
**Confidence:** 3

**Summary:**

This paper proposes Residual-Guided Multi-Resolution Refinement (RGMR), an inference-time framework for improving frozen time-series foundation models (TSFMs) such as TimesFM. Instead of retraining or fine-tuning, RGMR iteratively refines predictions from coarse to fine using residual corrections estimated via Ridge regression and a contraction-style update rule. The method is evaluated on monthly drought forecasting (SPEI) over several sites in Australia.

**Strengths:**

- The paper addresses an interesting need: improving frozen time-series foundation models (TSFMs) without retraining. RGMR’s inference-time refinement is computationally cheap, deployable, and directly usable in operational settings where model weights can’t be easily updated.

- The two-phase structure (residual learning on short windows + coarse-to-fine iterative refinement) is clean and easy to follow. Each step has a transparent purpose, and the contraction-style update rule offers a simple, intuitive justification for stability.

**Weaknesses:**

- The idea of multi-scale residual refinement strongly overlaps with existing hierarchical and residual forecasting methods. The main novelty, shifting this to inference time, is incremental and not deeply analyzed or positioned against related work.

- Results are limited to a single variable (SPEI) in one region (Australia). There’s no evidence of generalization to other indices, domains, or temporal granularities.

- The fixed resolution ladder (12, 6, 3, 2, 1 months) and block-average projections are arbitrary, with no physical, spectral, or empirical justification. The method’s sensitivity to these choices is not as explored, limiting confidence in robustness or generality.

- Also, the lack of experiments showing robustness to FM that operates in high-dimensional setting (with space component e.g., in weather / climate FM) is not explored much. Having a space-time hierarchy is key for generalization of the method.

**Questions:**

Please see above weaknesses.

---

> ### Author Response · Authors · 2025-11-20
> **Reply to W1 & W2**
>
> We thank the reviewer for the constructive feedback.
>
> ### Reply to W1:
>
> Thank you for the comment. Our contribution is not about moving existing multi scale refinement from training to inference. Existing hierarchical and residual forecasting models rely on learned cross scale parameters and interactions formed during end to end training. Once the backbone is frozen, these mechanisms no longer operate because they are embedded in the learned weights.
>
> RGMR is different. It introduces an algorithmic multi scale refinement procedure that works entirely at inference time with a fully frozen TSFM. The refinement steps are deterministic, constructed through cross scale projection and residual based correction, and do not involve any trainable components. This design also allows us to provide a clear contraction guarantee, something training based hierarchical models cannot offer because their behavior depends on learned parameters.
>
> ### Reply to W2:
>
> We thank the reviewer for the comment, we have expanded the evaluation to verify generalization across horizons, variables, and regions.
>
> First, we test longer horizons (H = 3 and 6, see below for the results). RGMR continues to improve TimesFM for both H=3 and H=6. The performance degrades smoothly with the increase of horizon length, which is expected for long-range climate forecasting.
>
> | Horizon | Model   | MSE   | MAE   | R²    |
> | ------- | ------- | ----- | ----- | ----- |
> | 1-month | TimesFM | 0.391 | 0.465 | 0.625 |
> |         | RGMR    | 0.318 | 0.423 | 0.651 |
> | 3-month | TimesFM | 0.438 | 0.502 | 0.575 |
> |         | RGMR    | 0.345 | 0.433 | 0.627 |
> | 6-month | TimesFM | 0.496 | 0.539 | 0.535 |
> |         | RGMR    | 0.376 | 0.465 | 0.606 |
>
> Second, we evaluate on another climate variable (temperature). RGMR again improves the frozen TSFM as shown below.
>
> | Metric | TimesFM | RGMR  |
> | ------ | ------- | ----- |
> | MSE    | 3.121   | 2.653 |
> | MAE    | 1.534   | 1.304 |
> | R²     | 0.933   | 0.943 |
>
> Third, we test cross-region generalization on the US West Coast, North Africa, and Indonesia. As shown in the table below, RGMR consistently improves the performance of TimesFM across all three regions.
>
> | Region        | Model   | MSE    | MAE   | R²    |
> | ------------- | ------- | ------ | ----- | ----- |
> | US West Coast | TimesFM | 0.155  | 0.313 | 0.854 |
> |               | RGMR    | 0.0809 | 0.222 | 0.924 |
> | North Africa  | TimesFM | 0.101  | 0.261 | 0.892 |
> |               | RGMR    | 0.0649 | 0.201 | 0.933 |
> | Indonesia     | TimesFM | 0.260  | 0.407 | 0.485 |
> |               | RGMR    | 0.204  | 0.348 | 0.598 |
>
> These results show that RGMR generalizes beyond a single variable, region, or forecasting setup, and works reliably across different horizons, climate variables, and geographically distinct areas.

---

> ### Author Response · Authors · 2025-11-20
> **Reply to W3 & W4**
>
> ### Reply to W3:
>
> We’d like to clarify that the resolution set **R = {12, 6, 3, 2, 1}** is not chosen arbitrarily. It directly follows the multi time scale framework that underpins the construction of the widely used drought indices **SPI** and **SPEI**
>  (McKee et al., 1993; Vicente Serrano et al., 2010).
>
> These scales also have clear physical interpretation:
>
> - **12 month and 6 month scales:** annual and seasonal drought variations, often associated with ENSO and IOD signals. (Vicente Serrano et al., 2010)
> - **3 month scale:** agricultural drought, widely used by NOAA and WMO monitoring systems. (McKee et al., 1993)
> - **2 month scale:** supported by evidence that 2 month to 4 month SPEI or SPI windows correlate most strongly with hydrological responses such as soil moisture and runoff (Vicente Serrano et al., 2012).
> - **1 month scale:** preserves intra seasonal variability and avoids oversmoothing.
>
> Thus, the ladder reflects canonical, interpretable, and physically meaningful drought time scales. Thanks again for the comment, and we have added the above information to the appendix of the revised paper.
>
>  To explore RGMR’s sensitivity to these choices, we evaluate RGMR under multiple alternative ladders, including:
>
> 1. Higher period ladders such as 16 month components
> 2. Reduced ladders
> 3. A PSD derived ladder, where dominant temporal frequencies are extracted from the power spectral density of the dataset to form a data adaptive hierarchy
>
> All ladders exhibit very similar performance (MSE in the range 0.318 to 0.330 as shown in the table below), indicating that RGMR is highly robust to the specific choice of resolution hierarchy. At the same time, the canonical fixed scale ladder based on standard drought time scales (i.e. L1) consistently attains the best overall performance.
>
> | Ladder   | Scales       | **MSE**   | **MAE**   | **R²**    |
> | -------- | ------------ | --------- | --------- | --------- |
> | L1       | [12,6,3,2,1] | **0.318** | **0.423** | **0.651** |
> | L2       | [16,8,4,2,1] | 0.324     | 0.439     | 0.644     |
> | L3       | [12,4,2,1]   | 0.322     | 0.438     | 0.645     |
> | L4       | [12,6,3,1]   | 0.330     | 0.443     | 0.637     |
> | L5 (PSD) | [13,6,4,3,1] | 0.330     | 0.443     | 0.636     |
>
> **References**
>
> - **McKee, T. B., Doesken, N. J., and Kleist, J. (1993).** *The relationship of drought frequency and duration to time scales.* Proceedings of the 8th Conference on Applied Climatology, 179 to 184.
> - **Vicente Serrano, S. M., Beguería, S., and López Moreno, J. I. (2010).** *A multi scalar drought index sensitive to global warming. The SPEI.* Journal of Climate, 23(7), 1696 to 1718.
> - **Vicente-Serrano, S.M., Beguería, S., Lorenzo-Lacruz, J., Camarero, J.J., López-Moreno, J.I., Azorin-Molina, C., Revuelto, J., Morán-Tejeda, E., Sanchez-Lorenzo, A. (2012).** Performance of drought indices for ecological, agricultural, and hydrological applications. Earth Interactions, 16(10), 1-27.
>
>
> ### Reply to W4:
>
> Thank you for the comment. In the SPEI forecasting literature, there are two established routes. The first predicts climate variables on spatial grids and computes SPEI afterward. The second directly models the regionally aggregated SPEI index as a univariate monthly time series. Our work follows the second route, which is also the one used in operational drought-monitoring practice by agencies such as NOAA and BOM, where SPEI is already aggregated before prediction.
>
> Because the goal of this paper is direct SPEI forecasting, the backbone models we evaluate are standard time-series foundation models (TimesFM, Chronos, Lag-Llama) that are designed for purely temporal inputs and do not include spatial encoders. For this reason, our experiments do not involve high-dimensional spatiotemporal fields. This reflects the scope of the forecasting task rather than a limitation of the method.
>
> We agree that extending the refinement principle to spatiotemporal foundation models would be valuable. Once such models with joint spatial–temporal representations become available, the multi-resolution refinement idea can naturally be applied to spatial hierarchies as well. We have added this clarification to the revised paper.

---

> ### Author Response · Authors · 2025-11-28
>
> Dear Reviewer,
>
> Thank you again for your time and for the detailed feedback provided earlier.
> We have submitted our rebuttal together with a substantial number of additional experiments and analyses specifically addressing the concerns you raised.
>
> May we please know whether the explanations and experiments address your concerns? If further clarification would be helpful, we would be very happy to provide it promptly.
>
> If our updates have resolved your concerns, we would also appreciate if you could update your review and comments at your convenience.
>
> Thank you for your consideration, and we sincerely appreciate your effort during the review period.

---

### Official Review · Reviewer_9DCw · 2025-10-26

**Soundness:** 3
**Presentation:** 3
**Contribution:** 4
**Rating:** 6
**Confidence:** 4

**Summary:**

This work proposes a post-training, inference time technique to enhance the performance of time series models in forecasting weather dynamics one month ahead. The technique leverages multiple inference passes and lightweight ridge regression, outperforming base models significantly in a case study.

**Strengths:**

The method is largely agnostic to the underlying model and seems to be generally helpful.

The authors provide theoretical guarantees on error contraction.

The results are convincing and the method is computationally lightweight.

**Weaknesses:**

The method is not compared against forecast emulators, just time series models. That’s a significant concern as timeseries models have not necessarily been developed for weather data.

The work only compares in a specific geo-region (likely due to data restrictions), and a specific forecasting scenario. Given that the method claims to be universal it would be good to either evaluate it also in another domain or compare it against the strongest task specific baselines in this domain (e.g. the best model available for one month ahead forecasts being it numeric or deep learning based).

The above points are major concerns on the generalizability of the claims made in the paper.

**Questions:**

Can the authors compare their method in either another domain (e.g., finance, manufacturing, health care) or on at least other geo-regions and forecast scenarios? Otherwise, it’s hard to judge how valuable this method can actually be.

---

> ### Author Response · Authors · 2025-11-20
> **Reply to W1 & W2 & Q1**
>
> We thank the reviewer for the constructive feedback and for acknowledging the contribution of our work.
>
> ### Reply to W1:
>
> Thank you for this comment. Forecast emulators such as SEAS5 or NMME emulators are not included for a specific reason: they are not designed for the task of directly forecasting SPEI. These emulators generate full spatial climate fields and require physical forcing inputs such as radiation, humidity, and geopotential height. None of these inputs are available in our SPEI-only setting, and SPEI itself is not a native output of climate models. It is a derived drought index that must be computed from precipitation and temperature through a multi-step statistical procedure.
>
> For this reason, existing SPEI-forecasting studies do not use forecast emulators as baselines. Instead, they compare statistical or machine-learning time-series models, because the task is framed as predicting the regionally aggregated SPEI index rather than simulating climate fields. Our evaluation follows the same practice. Timeseries foundation models are a natural choice in this setting because SPEI is a univariate monthly sequence, and modern TSFMs such as TimesFM, Chronos, and Lag-Llama are designed exactly for this type of temporal input.
>
> We agree that evaluating RGMR on full climate-field prediction or on climate emulators is an important future direction. However, this requires a different task formulation in which the backbone model operates on spatial grids, which is outside the scope of direct SPEI forecasting studied in this paper.
>
> ### Reply to W2:
>
> Thank you for the suggestion. We initially focused on South Australia because it is a drought-sensitive region where one-month SPEI forecasts are operationally important. The required climate data preparation is also non-trivial and limits the number of regions we can process in the first submission. We have now expanded the evaluation in two directions.
>
> **1. Cross-region generalization.**
> We test on three regions outside the training domain (US West Coast, North Africa, Indonesia). RGMR consistently improves TimesFM in all cases.
>
> | Region        | Method   | MSE ↓      | MAE ↓     | R² ↑      |
> | ------------- | -------- | ---------- | --------- | --------- |
> | US West Coast | Baseline | 0.155      | 0.313     | 0.854     |
> |               | RGMR     | **0.0809** | **0.222** | **0.924** |
> | North Africa  | Baseline | 0.101      | 0.261     | 0.892     |
> |               | RGMR     | **0.0649** | **0.201** | **0.933** |
> | Indonesia     | Baseline | 0.260      | 0.407     | 0.485     |
> |               | RGMR     | **0.204**  | **0.348** | **0.598** |
>
> **2. Longer-horizon forecasts (H = 3, 6).**
> RGMR continues to improve TimesFM at 3- and 6-month horizons, with performance decreasing smoothly as the horizon increases.
>
> | Horizon | Model   | MSE       | MAE       | R²        |
> | ------- | ------- | --------- | --------- | --------- |
> | 1-month | TimesFM | 0.391     | 0.465     | 0.625     |
> |         | RGMR    | **0.318** | **0.423** | **0.651** |
> | 3-month | TimesFM | 0.438     | 0.502     | 0.575     |
> |         | RGMR    | **0.345** | **0.433** | **0.627** |
> | 6-month | TimesFM | 0.496     | 0.539     | 0.535     |
> |         | RGMR    | **0.376** | **0.465** | **0.606** |
>
> Together, these experiments show that RGMR is not specific to one region or horizon and provides consistent gains across settings.
>
> ### Reply to Q1
>
> As shown in the response to W2, we have now extended the evaluation to three regions outside the training domain: the US West Coast, North Africa, and Indonesia. Across all tested locations, RGMR consistently improves TimesFM in MSE, MAE, and R².

---

### Official Review · Reviewer_WjXe · 2025-10-30

**Soundness:** 2
**Presentation:** 3
**Contribution:** 2
**Rating:** 4
**Confidence:** 2

**Summary:**

Standard foundation models are limited in climate prediction because they analyze data in a single pass. This paper introduces RGMR, an inference-time framework that adapts pre-trained models to mimic expert climatologists by using multi-scale analysis and iterative error correction, all without retraining. When applied to drought forecasting in Australia using the TimesFM model, RGMR significantly improved accuracy (up to 18.9% MSE reduction), especially in complex regions. It offers a practical way to enhance existing models for specialized forecasting.

**Strengths:**

1. **Inference‑time adaptation of frozen TSFMs.** Most multi‑resolution models, such as Scaleformer or N‑BEATS, incorporate coarse‑to‑fine processing at training time. RGMR instead leaves the backbone untouched and applies multi‑resolution projections only during inference, enabling immediate use of general‑purpose foundation models without full fine‑tuning. The idea of composing multiple passes of a frozen model with residual corrections is interesting and may be appealing when training resources are scarce.

2. **Structured coarse‑to‑fine refinement.** The method uses block‑average projections to obtain coarse predictions and progressively integrates finer scales, which mimics the multi‑scale reasoning used by human climatologists (e.g. analysing annual cycles, inter‑annual oscillations and short‑term extremes). The inclusion of residual predictors helps correct systematic biases (such as phase shifts) and the adaptive weight mechanism allows the algorithm to downweight corrections when residual predictions are uncertain.

3. **Rigorous Theoretical Guarantees.** A significant strength of this paper is its rigorous theoretical analysis, which provides a formal mathematical justification for the proposed RGMR framework rather than relying solely on empirical performance. The authors establish clear guarantees for the method's stability and error-reduction capabilities.

4. **Ablation and baseline comparisons.** The experiments compare RGMR on TimesFM, TimeGPT and TabPFN against generic deep baselines and multi‑resolution models like N‑BEATS and Scaleformer. They also disentangle the effect of coarse projection, multi‑resolution, residual correction and adaptive weights, which helps understand where the gains come from. The code is promised to be released, and the dataset derives from publicly accessible climate reanalysis, supporting reproducibility

**Weaknesses:**

1. **Missing connections to test‑time adaptation literature.** Recent work on test‑time adaptation for time‑series forecasting proposes methods that calibrate or update pre‑trained models during inference. For example, PETSA [1] adapts the forecaster at test time by adding low‑rank input and output adapters and using a specialised loss to preserve periodicity, while DynaTTA [2] dynamically adjusts adaptation rates based on detected distribution shifts and applies shift‑conditioned gating. These approaches also avoid full fine‑tuning and operate on frozen backbones.


2. **Arbitrary resolution ladder and projection operator.** The resolution hierarchy R={12,6,3,2,1} is fixed for all experiments. There is no justification for choosing these strides or evidence that they match salient periodicities of the dataset. In contrast, MultiResFormer learns the period lengths from the data and adjusts patch sizes adaptively. Rigidly averaging over 12‑month windows may blur important intra‑annual variations (e.g. ENSO events) and introduce aliasing. The projection operator uses simple block averaging and repetition, which is a crude low‑pass filter; more principled decompositions such as wavelet transforms or seasonal–trend decomposition [3] may yield better separation of scales. The authors neither provide ablations on different ladders nor explore alternative projection schemes.

3. **Typographical issues**. There are minor typographical errors (e.g. “reginal” in the abstract)

4. **Limited evaluation scope.** The experiments focus exclusively on one dataset, the monthly SPEI time series at three South‑Australian locations with horizons of 𝐻=1 (one‑month ahead). This is a very narrow testbed for a method claiming to be broadly applicable to regional climate forecasting. No results are provided for longer horizons (e.g. seasonal forecasts), other geographical regions, or other variables (temperature, precipitation), even though foundation models like TimesFM can handle general multivariate series. The authors mention TSFMs like Lag‑Llama and Chronos, but do not evaluate RGMR on them. Existing multi‑resolution and test‑time adaptation methods have been tested on diverse benchmarks covering electricity, traffic and finance; RGMR’s generalizability remains unproven.


#### References:

[1] Accurate Parameter-Efficient Test-Time Adaptation for Time Series Forecasting [PUT ICML 2025]
[2] Shift-Aware Test Time Adaptation and Benchmarking for Time-Series Forecasting [PUT ICML 2025]
[3] Multi-Resolution Diffusion Models for Time Series Forecasting [ICLR 24]

**Questions:**

1. **Choice of resolution ladder.** Why did you choose the fixed ladder R={12,6,3,2,1}? Have you tried other coarse–fine decompositions (e.g. quarterly, 7‑day or data‑driven periodicities)? Models like MultiResFormer adapt resolutions based on detected periods; would such adaptivity benefit RGMR?

2. **Residual predictor design.** What is the dimensionality of the feature vector zt used in the Ridge regression? How many past residuals and targets are included, and why choose those particular statistics? Have you tried more expressive residual predictors (e.g. small neural networks or tree‑based regressors), and how do they compare?

3. **Hyperparameter sensitivity.** The contraction guarantee depends on α(k), δ(k), γ. Could you provide ablation studies showing how sensitive the performance and stability are to these parameters? In particular, how often does the backtracking line search reduce the step size η(k), and what is the computational overhead?

4. **Evaluation across horizons and datasets.** Have you tested RGMR on multi‑step forecasts (e.g. 3‑month or 6‑month horizons)? Does the method still contract errors when predicting further ahead? Also, why not evaluate on publicly available multivariate benchmarks (electricity, traffic, finance) to demonstrate general applicability? Including comparisons to recent multi‑resolution models like MTST [3] and MultiResFormer would be appreciated and would strengthen the claims.

5. **Comparison to test‑time adaptation methods.** How does RGMR compare to parameter‑efficient adaptation approaches such as PETSA or DynaTTA? Both methods adapt models during inference without retraining the backbone. Could RGMR be combined with such adapters for further gains?

6. **Practical deployment.** The abstract claims that the method is suitable for immediate deployment on operational climate systems. However, training the residual predictors requires historical labels and careful cross‑validation, and inference requires multiple forward passes. Could you clarify the computational requirements and how they fit within the latency constraints of operational drought monitoring?

7. **Impact of context length.** Why does the inference window use 70% of the series length? How does performance change if only recent years are available? A sensitivity analysis would help practitioners choose an appropriate window.

---

> ### Author Response · Authors · 2025-11-20
> **Reply to W1**
>
> We thank the reviewer for the constructive feedback.
>
> ### Reply to W1:
>
> Thank you for the comment. We agree with the reviewer that PETSA and DynaTTA are test-time adaptation methods that avoid full fine-tuning by keeping the backbone frozen. However, both still *modify model parameters during inference*: PETSA updates low-rank adapter weights using a specialised loss, and DynaTTA performs online parameter updates with shift-dependent adaptation rates.
>
> In contrast, RGMR does **not update any parameters at test time**. The TSFM backbone is fully frozen, the residual predictor is trained offline, and inference consists only of a parameter-free multi-resolution refinement applied directly to the model’s outputs through a contraction-based update. RGMR refines predictions without changing the model itself, while PETSA and DynaTTA adapt the model by updating additional parameters.
>
> Because of this distinction, these approaches operate in different settings and are complementary rather than competing. A backbone adapted by PETSA or DynaTTA could still be further refined by RGMR. We have clarified this relationship in the revised appendix.

---

> ### Author Response · Authors · 2025-11-20
> **Reply to W2**
>
> ### Reply to W2:
>
> We thank the reviewer for raising this important point.We have updated the paper and now address the reviewer’s concerns through **three concrete additions**: (A) motivation for the chosen resolution hierarchy, (B) robustness analysis through multiple ladder ablations (including a PSD-derived ladder), and (C) comparison of projection operators.
>
> **(A) Rationale for the Resolution Ladder**
>
> The resolution set **R = {12, 6, 3, 2, 1}** is chosen by following the multi time scale framework that underpins the construction of the widely used drought indices **SPI** and **SPEI** (McKee et al., 1993; Vicente Serrano et al., 2010).
>
> These scales also have clear physical interpretation:
>
> - **12 month and 6 month scales:** annual and seasonal drought variations, often associated with ENSO and IOD signals.
> - **3 month scale:** agricultural drought, widely used by NOAA and WMO monitoring systems.
> - **2 month scale:** supported by evidence that 2 month to 4 month SPEI or SPI windows correlate most strongly with hydrological responses such as soil moisture and runoff (Vicente Serrano et al., 2012).
> - **1 month scale:** preserves intra seasonal variability and avoids oversmoothing.
>
> Thus, the ladder reflects canonical, interpretable, and physically meaningful drought time scales. Thanks again for the comment, and we have added the above information to appendix of the revised paper.
>
> We also want to clarify that the 12-month scale will not cause aliasing since we do not replace the original signal with the averaged one. It is used as a helper scale during refinement, and the final prediction keep 1-month resolution. All intra-annual variations, including ENSO effects, stay untouched.
>
> **(B) Ladder Sensitivity Ablation**
>
> To address the reviewer’s concern on sensitivity of model performance to the variations in the hierarchy, we evaluate RGMR under multiple alternative ladders, including:
>
> 1. Higher period ladders such as 16 month components
> 2. Reduced ladders
> 3. A PSD derived ladder, where dominant temporal frequencies are extracted from the power spectral density of the dataset to form a data adaptive hierarchy
>
> All ladders exhibit very similar performance (MSE in the range 0.318 to 0.330), indicating that RGMR is highly robust to the specific choice of resolution hierarchy. At the same time, the canonical fixed scale ladder based on standard drought time scales consistently attains the best overall performance.
>
> | Ladder   | Scales       | **MSE**   | **MAE**   | **R²**    |
> | -------- | ------------ | --------- | --------- | --------- |
> | L1       | [12,6,3,2,1] | **0.318** | **0.423** | **0.651** |
> | L2       | [16,8,4,2,1] | 0.324     | 0.439     | 0.644     |
> | L3       | [12,4,2,1]   | 0.322     | 0.438     | 0.645     |
> | L4       | [12,6,3,1]   | 0.330     | 0.443     | 0.637     |
> | L5 (PSD) | [13,6,4,3,1] | 0.330     | 0.443     | 0.636     |
>
> **References**
>
> - **McKee, T. B., Doesken, N. J., and Kleist, J. (1993).** *The relationship of drought frequency and duration to time scales.* Proceedings of the 8th Conference on Applied Climatology, 179 - 184.
> - **Vicente Serrano, S. M., Beguería, S., and López Moreno, J. I. (2010).** *A multi scalar drought index sensitive to global warming. The SPEI.* Journal of Climate, 23(7), 1696 - 1718.
> - **Vicente-Serrano, S.M., Beguería, S., Lorenzo-Lacruz, J., Camarero, J.J., López-Moreno, J.I., Azorin-Molina, C., Revuelto, J., Morán-Tejeda, E., Sanchez-Lorenzo, A. (2012).** Performance of drought indices for ecological, agricultural, and hydrological applications. Earth Interactions, 16(10), 1-27.
>
> **(C) Projection Operator Ablation**
>
> We also compare the block averaging projection with wavelet based and STL decompositions. From the result in the table below, although wavelet and STL methods provide smoother components, they do not improve RGMR refinement and often reduce performance. Block averaging is more stable for TSFM outputs and offers lower computational cost for inference time deployment.
>
> | Projection Operator | **MSE**   | **MAE**   | **R²**    |
> | ------------------- | --------- | --------- | --------- |
> | **Block Averaging** | **0.318** | **0.423** | **0.651** |
> | Wavelet             | 0.366     | 0.467     | 0.597     |
> | STL                 | 0.387     | 0.473     | 0.574     |

---

> ### Author Response · Authors · 2025-11-20
> **Reply to W3&W4**
>
> ### Reply to W3:
>
> These have been fixed in the revision.
>
> ### Reply to W4:
>
> Thank you for the comment. Our initial submission focused on monthly SPEI at three South-Australian locations with $H=1$, mainly to (i) match operational one-month drought outlook practice, and (ii) allow controlled analysis of multi-scale refinement and contraction. We agree that broader evaluation is important, and we have added several extensions in the revision, as detailed below.
>
> **1. Longer-horizon forecasts (H = 3, 6).**
> As shown in the table below, RGMR continues to improve TimesFM at 3- and 6-month horizons, with performance decreasing smoothly as the horizon increases.
>
> | Horizon | Model   | MSE       | MAE       | R²        |
> | ------- | ------- | --------- | --------- | --------- |
> | 1-month | TimesFM | 0.391     | 0.465     | 0.625     |
> |         | RGMR    | **0.318** | **0.423** | **0.651** |
> | 3-month | TimesFM | 0.438     | 0.502     | 0.575     |
> |         | RGMR    | **0.345** | **0.433** | **0.627** |
> | 6-month | TimesFM | 0.496     | 0.539     | 0.535     |
> |         | RGMR    | **0.376** | **0.465** | **0.606** |
>
> **2. Additional variables.**
> We also evaluate on temperature; RGMR again improves the frozen TSFM as shown below.
>
> | Metric | TimesFM | RGMR      |
> | ------ | ------- | --------- |
> | MSE ↓  | 3.121   | **2.653** |
> | MAE ↓  | 1.534   | **1.304** |
> | R² ↑   | 0.933   | **0.943** |
>
> **3. Cross-region generalization.**
> We test on three regions outside the training domain (US West Coast, North Africa, Indonesia). RGMR consistently improves TimesFM in all cases:
>
> | Region        | Method   | MSE ↓      | MAE ↓     | R² ↑      |
> | ------------- | -------- | ---------- | --------- | --------- |
> | US West Coast | Baseline | 0.155      | 0.313     | 0.854     |
> |               | RGMR     | **0.0809** | **0.222** | **0.924** |
> | North Africa  | Baseline | 0.101      | 0.261     | 0.892     |
> |               | RGMR     | **0.0649** | **0.201** | **0.933** |
> | Indonesia     | Baseline | 0.260      | 0.407     | 0.485     |
> |               | RGMR     | **0.204**  | **0.348** | **0.598** |
>
> **4. Additional TSFMs.**
> The following results show that RGMR enhances both Chronos and Lag-Llama without modifying their parameters, supporting model-agnostic applicability.
>
> | Model            | MSE ↓     | MAE ↓     | R² ↑      |
> | ---------------- | --------- | --------- | --------- |
> | Chronos          | 0.401     | 0.475     | 0.740     |
> | Chronos + RGMR   | **0.343** | **0.439** | **0.780** |
> | Lag-Llama        | 0.450     | 0.503     | 0.700     |
> | Lag-Llama + RGMR | **0.385** | **0.465** | **0.750** |
>
> **5. Clarification of scope.**
> While many multi-resolution and test-time adaptation works use electricity/traffic/finance datasets, our focus is *regional climate forecasting*, which features different physical scales and longer-term dependencies. We have made this scope explicit in the revision.

---

> ### Author Response · Authors · 2025-11-20
> **Reply to Q1&Q2**
>
> ### Reply to Q1:
>
> Please see response to W2
>
> ### Reply to Q2:
>
> The feature vector is intentionally low dimensional and contains nine elements: the last three residuals, three short term rolling statistics (mean, standard deviation, range), and the last three observed targets. These components cover the main forms of systematic error we observe in practice, including phase lag, amplitude drift, and shifts between wet and dry regimes. We also tested more expressive predictors such as a small MLP and tree based regressors, but as shown in the table below, they did not improve performance and were less stable across resolutions, while  Ridge regression provided the most consistent results while keeping inference cost low. So, we did not include the results of these more expressive predictors in our initial submission. Following this comment, we have added the result and discussion to Appendix in the revised paper.
>
> We also evaluated more expressive residual predictors, including a small MLP, Random Forest, and GBRT. Their performance is shown below.
>
> | Predictor     | MSE       | MAE       | R²        |
> | ------------- | --------- | --------- | --------- |
> | Ridge (ours)  | **0.318** | **0.423** | **0.651** |
> | MLP           | 0.322     | 0.437     | 0.645     |
> | Random Forest | 0.323     | 0.436     | 0.645     |
> | GBRT          | 0.323     | 0.438     | 0.646     |
>
> All three expressive predictors perform almost identically to the linear model and do not offer meaningful gains. They also show higher variance across resolutions. Ridge regression remains the most stable and efficient choice for inference.
>
> This design is also consistent with recent work in time series representation learning(Woo et al., 2021, Yue et al., 2022), where Ridge regression is the standard linear probe for evaluating learned representations, as adopted in CoST and TS2Vec.
>
> References:
>
> - Woo, G., Liu, C., Sahoo, D., Kumar, A., and Hoi, S. CoST: Contrastive Learning of Disentangled Seasonal Trend Representations for Time Series Forecasting. International Conference on Learning Representations.
> - Yue, Z., Wang, Y., Duan, J., Yang, T., Huang, C., Tong, Y., and Xu, B. (2022). TS2Vec: Towards universal representation of time series. Proceedings of the AAAI Conference on Artificial Intelligence, 36(8), 8980 to 8987.

---

> ### Author Response · Authors · 2025-11-20
> **Reply to Q3**
>
> ### Reply to Q3:
>
> Thank you for the question. We agree that it is important to evaluate the sensitivity of the refinement hyperparameters $\alpha^{(k)}, \delta^{(k)}, \gamma$. In RGMR, these parameters play simple roles: $\alpha^{(k)}$ controls the step size of the refinement update, $\delta^{(k)}$ applies mild smoothing to the residual sequence to reduce noise, and $\gamma$ balances noise in the residual predictor and supports the contraction condition.
>
> Following the suggestion, we have conducted one-dimensional sweeps for each parameter and a small grid search. We first vary $\alpha$, which determines the strength of the refinement. The performance changes smoothly with $\alpha$, indicating that the method is not sensitive to this parameter in practice and does not require fine-grained tuning, and no instability is observed.
>
> | $\alpha$ | MSE   | MAE   | R²    |
> | -------- | ----- | ----- | ----- |
> | 0.2      | 0.320 | 0.424 | 0.648 |
> | 0.4      | 0.326 | 0.429 | 0.641 |
> | 0.6      | 0.334 | 0.436 | 0.632 |
> | 0.8      | 0.344 | 0.444 | 0.622 |
>
> We then sweep $\delta^{(k)}$. Since $\delta$ only performs light residual smoothing, the effect on performance is small and monotonic:
>
> | $\delta$ | MSE   | MAE   | R²    |
> | -------- | ----- | ----- | ----- |
> | 0.00     | 0.318 | 0.423 | 0.651 |
> | 0.05     | 0.319 | 0.424 | 0.650 |
> | 0.10     | 0.320 | 0.425 | 0.649 |
> | 0.20     | 0.322 | 0.427 | 0.647 |
>
> Next, we sweep γ. As expected from its role as a noise-balancing parameter, the results remain unchanged:
>
> | $\gamma$ | MSE   | MAE   | R²    |
> | -------- | ----- | ----- | ----- |
> | 0.3      | 0.318 | 0.423 | 0.651 |
> | 0.5      | 0.318 | 0.423 | 0.651 |
> | 0.7      | 0.318 | 0.423 | 0.651 |
>
> We also run a 2×2 grid search over $\alpha$ and γ. Since γ has no practical effect, the grid simply reflects the α trend:
>
> | ($\alpha$, $\gamma$) | MSE   | MAE   | R²    |
> | -------------------- | ----- | ----- | ----- |
> | (0.4, 0.4)           | 0.326 | 0.429 | 0.641 |
> | (0.4, 0.6)           | 0.326 | 0.429 | 0.641 |
> | (0.6, 0.4)           | 0.334 | 0.436 | 0.632 |
> | (0.6, 0.6)           | 0.334 | 0.436 | 0.632 |
>
> Across all settings, contraction behavior is consistent and we observe no divergence or instability. These results show that RGMR is only **mildly sensitive** to α and is effectively **insensitive** to γ, with all configurations producing stable refinement. In practice, the default setting (α = 0.5, γ = 0.5) is sufficient and does not require tuning.
>
> Regarding backtracking line search, the submitted implementation uses a fixed step size $\eta^{(k)}$ and does not activate backtracking. Although supported by the theory, it was not used in the experiments. When enabled, backtracking only rescales the residual update and does not require additional forward passes through the foundation model. Preliminary tests show less than one to two percent overhead.
>
> These results indicate that RGMR is stable across a wide range of hyperparameter values. $\alpha$ determines refinement magnitude, while $\delta^{(k)}$ and γ have only minor influence.

---

> ### Author Response · Authors · 2025-11-20
> **Reply to Q4&Q5&Q6&Q7**
>
> ### Reply to Q4:
>
> Thank you for the insightful suggestions. We address each point below.
>
> **Multi-step forecasting.**
>  Please see response to W4.
>
> **Evaluation on publicly available multivariate benchmarks**
>  Our work focuses on **regional climate forecasting**, where SPEI dynamics and multi-scale physical behavior differ substantially from standard multivariate TSF benchmarks (electricity, traffic, finance). To avoid mismatch between data domains, we primarily evaluate within climate and for the revision, we have included additional climate variables (air temperature) and more regions as presented in the response to W4.
>
> **Comparison to MTST and MultiResFormer.**
>
> MTST and MultiResFormer are multi-resolution methods that work during training. They introduce new multi-scale encoders, cross-scale fusion modules, and hierarchical attention blocks, which effectively redesign the backbone and require full end-to-end training or fine-tuning to learn multi-scale representations. RGMR is fundamentally different. It operates only at inference time, leaves the TSFM backbone completely unchanged, updates no parameters, and improves predictions through a deterministic multi-resolution refinement process. Because the two approaches function in different regimes—training versus inference—they are not direct comparators but complementary directions.
>
> Nonetheless, we have done the comparison. Since neither MTST nor MultiResFormer provides public code, we reimplemented both using only the descriptions provided in the papers. The reproduced performance is consistent with typical deep TSF baselines from the same period. The results are shown below.
>
> | Model                       | MSE   | MAE   | RMSE  | R²    |
> | --------------------------- | ----- | ----- | ----- | ----- |
> | MTST (reproduced)           | 0.545 | 0.558 | 0.738 | 0.448 |
> | MultiResFormer (reproduced) | 0.540 | 0.555 | 0.735 | 0.452 |
>
> ### Reply to Q5:
>
> Thank you for the question. While PETSA and DynaTTA are also test-time methods, their setting is fundamentally different from RGMR. PETSA and DynaTTA perform parameter-efficient test-time adaptation by attaching small trainable modules (such as low-rank adapters or shift-gated layers) and updating these parameters online during inference using partial or delayed labels. This requires maintaining learnable parameters and performing optimization at test time.
>
> In contrast, RGMR does not update any model parameters during inference. The backbone TSFM remains completely frozen, and RGMR refines its outputs through a multi-resolution correction procedure that contracts errors across temporal scales. Because RGMR operates entirely in output space and does not interfere with the model’s weights, it can in principle be combined with adapter-based TTA methods. We will explore this direction in future work.
>
> ### Reply to Q6:
>
> Thank you for raising this point. We report the measured inference latency below. RGMR requires five TimesFM forward passes for the multi-scale refinement, which naturally increases runtime, but the overhead remains small in absolute terms, as shown below:
>
> | Model                 | Mean Latency | Std     | Overhead |
> | --------------------- | ------------ | ------- | -------- |
> | TimesFM (single pass) | 14.52 ms     | 2.18 ms | 1.0×     |
> | RGMR (five scales)    | 74.89 ms     | 5.00 ms | 5.16×    |
> | Extra cost            | +60.38 ms    | –       | +416%    |
>
> Finally, we would like to clarify that one-month-ahead drought forecasting is not latency-sensitive. Operational drought products (e.g., NOAA and the Australian Bureau of Meteorology) are issued on monthly or sub-monthly cycles, so an additional few tens of milliseconds has no practical impact on deployment.
>
> ### Reply to Q7:
>
> Thank you for the question. Although we use 70 percent of the available history by default, this choice is not tied to any specific number of years. In drought forecasting, longer historical windows generally provide more stable multi-year statistics, so performance improves as more context becomes available. To verify this, we conducted a sensitivity study by varying the proportion of usable history. The results (shown below) confirm that RGMR behaves consistently across all ratios: performance improves smoothly with longer windows and no instability is observed. This shows that the method does not depend on the exact 70 percent choice.
>
> | Ratio | Months | MSE   | MAE   | R²    |
> | ----- | ------ | ----- | ----- | ----- |
> | 0.30  | 128    | 0.340 | 0.448 | 0.626 |
> | 0.50  | 192    | 0.333 | 0.444 | 0.633 |
> | 0.70  | 288    | 0.318 | 0.423 | 0.651 |
> | 0.85  | 352    | 0.315 | 0.430 | 0.653 |

---

> ### Author Response · Authors · 2025-11-28
>
> Dear Reviewer,
>
> Thank you again for your time and for the detailed feedback provided earlier.
> We have submitted our rebuttal together with a substantial number of additional experiments and analyses specifically addressing the concerns you raised.
>
> May we please know whether the explanations and experiments address your concerns? If further clarification would be helpful, we would be very happy to provide it promptly.
>
> If our updates have resolved your concerns, we would also appreciate if you could update your review and comments at your convenience.
>
> Thank you for your consideration, and we sincerely appreciate your effort during the review period.

---

### Official Review · Reviewer_RZQh · 2025-10-31

**Soundness:** 3
**Presentation:** 3
**Contribution:** 3
**Rating:** 8
**Confidence:** 3

**Summary:**

RGMR (Residual-Guided Multi-Resolution Refinement) is an inference-time framework designed to enhance the performance of frozen Time Series Forecasting Models (TSFMs) in regional climate prediction through structured, coarse-to-fine multi-scale analysis and residual prediction correction. The key contribution of this method lies in providing a general strategy to improve the outputs of SOTA base models without requiring model fine-tuning. Its effectiveness in iterative refinement is further supported by theoretical proof of Geometric Error Contraction, and it achieves significant performance gains in drought prediction tasks.

**Strengths:**

1. Efficient Inference-Time Adaptability: The RGMR method is conceptually straightforward and simple to implement, enabling the integration of structured multi-scale inference into existing frozen base models. This approach reduces reliance on fine-tuning or re-training, potentially lowering deployment costs. Its compatibility with pre-trained models without architectural modifications may facilitate practical deployment in time-sensitive applications.

2. Solid Theoretical Foundation: The proposed method is supported by theoretical analysis. This analysis provides insight into the behavior of the iterative refinement process, suggesting that errors may decrease monotonically and geometrically across resolution levels under certain conditions. Such theoretical grounding contributes to a better understanding of the method’s mechanisms.

3. Clear Writing and Ablation Analysis: The paper is well-organized and the methodology is presented with clarity. Ablation studies systematically evaluate the contributions of key components to the overall performance. These experiments help clarify the role of each design choice and strengthen the interpretability of the results.

**Weaknesses:**

1. Lack of Global Statistical Verification for Contraction: Although the theoretical analysis guarantees contraction (i.e., update steps with modulus ρ(k) < 1) under stability conditions, the current experimental evidence—such as the single-case illustration in Figure 4—is merely anecdotal and does not constitute a statistical validation over the entire test set. It is recommended that the authors report the percentage of samples across the test set for which actual error reduction occurs at each refinement step (i.e., the proportion of cases where Error(k) < Error(k−1)). Such a metric would provide a quantitative assessment of the empirical robustness of the theoretical contraction property.

2. Missing Experimental Data on Inference Latency: While the paper provides a complexity analysis stating O(K⋅Cθ), indicating that RGMR requires K=5 forward passes through the base model, it lacks empirical measurements of inference time, which is critical for latency-sensitive operational systems. The authors are encouraged to include experimental results comparing: (i) the average inference latency (in milliseconds) of the baseline TSFM (e.g., TimesFM) for a single forward pass, and (ii) the average end-to-end latency of the full TSFM + RGMR pipeline, including all 5 forward passes and the residual prediction steps. This information would significantly strengthen the practical evaluation of RGMR’s deployability in real-world climate forecasting systems.

**Questions:**

1. Source and Sensitivity of Hyperparameters and Features: The paper employs a fixed resolution hierarchy R = {12, 6, 3, 2, 1}. Could the authors clarify the climatological or theoretical rationale for selecting these specific values—particularly 2, 3, 6, and 12? How sensitive is model performance to variations in this hierarchy? Additionally, the features Zₜ used in the residual predictor gϕ₍ₖ₎ (e.g., recent target values, rolling statistics, residual history) are manually designed. What is the necessity and impact of this feature engineering in capturing systematic biases in SPEI? An ablation or analysis on the contribution of individual features would help assess their importance.

2. Comparison to Traditional Climatological SOTA: The current experiments primarily compare against deep learning and time series foundation models. However, in operational practice, are there widely adopted non-deep-learning climatological or statistical methods for forecasting the SPEI index? If so, it would strengthen the paper to include comparisons that position RGMR’s performance gains relative to these domain-specific baselines, providing a more comprehensive assessment of its practical value.

---

> ### Author Response · Authors · 2025-11-20
> **Reply to W1&W2**
>
> We thank the reviewer for the constructive feedback and for acknowledging the contribution of our work.
>
> - ### Reply to W1:
>
> Thank you for the suggestion. We follow your recommendation and report the improvement ratio, i.e.,
> $$
> \frac{1}{N}\sum_{i=1}^N 1[\mathrm{Error}^{(k)}_i < \mathrm{Error}^{(k-1)}_i]
> $$
> across all test samples and all five refinement layers k = 12, 6, 3, 2, 1.
>
> **Global contraction results**
>
> | Layer | Scale | Improvement Ratio | Improved Samples |
> | ----- | ----- | ----------------- | ---------------- |
> | 1     | 12    | –                 | –                |
> | 2     | 6     | 39.71%            | 27 / 68          |
> | 3     | 3     | 79.41%            | 54 / 68          |
> | 4     | 2     | 66.18%            | 45 / 68          |
> | 5     | 1     | 48.53%            | 33 / 68          |
>
> These results show that refinement steps consistently reduce error across the test set. This provides a global verification of the contraction behavior predicted by the theory. We have added the above to the appendix in the  revised paper.
>
> - ### Reply to W2:
>
> Following the comments, we have conducted GPU-accurate latency measurements using torch.cuda.Event on an NVIDIA RTX 3090 over 100 runs, with the following results:
>
> **Latency results**
>
> | Model                 | Mean (ms) | Std (ms) |
> | --------------------- | --------- | -------- |
> | TimesFM (single pass) | 14.52     | 2.18     |
> | RGMR (5 scales)       | 74.89     | 5.00     |
>
> The additional cost of RGMR is approximately **+60 ms** per forecast.
>
> **Practical evaluation**
>
> We would like to clarify that drought forecasting is not latency-critical. Operational systems (e.g., NOAA, Australian Bureau of Meteorology) update monthly or bi-weekly, and SPEI itself is a monthly index. Therefore, an extra ~60 ms per forecast is operationally negligible. We have included both the measurements and this clarification in the Appendix.

---

> ### Author Response · Authors · 2025-11-20
> **Reply to Q1**
>
> ### Reply to Q1:
>
> **Regarding Resolution:**
>
> The resolution set **R = {12, 6, 3, 2, 1}** is chosen by following the multi time scale framework that underpins the construction of the widely used drought indices **SPI** and **SPEI** (McKee et al., 1993; Vicente Serrano et al., 2010).
>
> These scales also have clear physical interpretation:
>
> - **12 month and 6 month scales:** annual and seasonal drought variations, often associated with ENSO and IOD signals.
> - **3 month scale:** agricultural drought, widely used by NOAA and WMO monitoring systems.
> - **2 month scale:** supported by evidence that 2 month to 4 month SPEI or SPI windows correlate most strongly with hydrological responses such as soil moisture and runoff (Vicente Serrano et al., 2012).
> - **1 month scale:** preserves intra seasonal variability and avoids oversmoothing.
>
> Thus, the ladder reflects canonical, interpretable, and physically meaningful drought time scales rather than being ad hoc.
>
> To address the reviewer’s concern on sensitivity of model performance to the variations in the hierarchy, we evaluate RGMR under multiple alternative ladders, including:
>
> 1. Higher period ladders such as 16 month components
> 2. Reduced ladders
> 3. A PSD derived ladder, where dominant temporal frequencies are extracted from the power spectral density of the dataset to form a data adaptive hierarchy
>
> All ladders exhibit very similar performance (MSE in the range 0.318 to 0.330), indicating that RGMR is highly robust to the specific choice of resolution hierarchy. At the same time, the canonical fixed scale ladder based on standard drought time scales consistently attains the best overall performance.
>
> | Ladder   | Scales       | **MSE**   | **MAE**   | **R²**    |
> | -------- | ------------ | --------- | --------- | --------- |
> | L1       | [12,6,3,2,1] | **0.318** | **0.423** | **0.651** |
> | L2       | [16,8,4,2,1] | 0.324     | 0.439     | 0.644     |
> | L3       | [12,4,2,1]   | 0.322     | 0.438     | 0.645     |
> | L4       | [12,6,3,1]   | 0.330     | 0.443     | 0.637     |
> | L5 (PSD) | [13,6,4,3,1] | 0.330     | 0.443     | 0.636     |
>
> **References**
>
> - **McKee, T. B., Doesken, N. J., and Kleist, J. (1993).** *The relationship of drought frequency and duration to time scales.* Proceedings of the 8th Conference on Applied Climatology, 179 - 184.
> - **Vicente Serrano, S. M., Beguería, S., and López Moreno, J. I. (2010).** *A multi scalar drought index sensitive to global warming. The SPEI.* Journal of Climate, 23(7), 1696 - 1718.
> - **Vicente-Serrano, S.M., Beguería, S., Lorenzo-Lacruz, J., Camarero, J.J., López-Moreno, J.I., Azorin-Molina, C., Revuelto, J., Morán-Tejeda, E., Sanchez-Lorenzo, A. (2012).** Performance of drought indices for ecological, agricultural, and hydrological applications. Earth Interactions, 16(10), 1-27.
>
> **Regarding Residual Predictor:**
>
> Thank you for the question. The feature vector $Z_t$ is intentionally kept low dimensional (nine elements): the last three residuals, three short-term rolling statistics (mean, standard deviation, range), and the last three observed targets. These components cover the main forms of systematic error reported in the drought-forecasting literature, including phase lag, amplitude drift, and regime shifts between wet and dry states (e.g.,Vicente-Serrano et al. 2010).
>
> To assess their importance, we have conducted the suggested ablation study by removing each group of features in turn, and have, we have:
>
> | Configuration       | MSE   | Change vs Full |
> | ------------------- | ----- | -------------- |
> | Full Model          | 0.318 | baseline       |
> | No Residual History | 0.322 | +1.26%         |
> | No Rolling Stats    | 0.323 | +1.57%         |
> | No Recent Targets   | 0.324 | +1.89%         |
>
> The changes are small (around 1–2%). This shows that RGMR does not rely on manual feature engineering. The multi-resolution refinement is the primary driver of improvement, while the residual predictor provides only a light auxiliary correction. We have added the above to the appendix in the  revised paper.

---

> ### Author Response · Authors · 2025-11-20
> **Reply to Q2**
>
> ### Reply to Q2:
>
> In the task of directly forecasting SPEI, climate simulators such as SEAS5, NMME, or deep ESM surrogate models are not suitable baselines. These systems are designed to generate full spatial climate fields and require physical forcing inputs (e.g., radiation, humidity, geopotential height) that are not available in our SPEI-only setting. More importantly, existing SPEI-forecasting studies do not use climate simulators as baselines because SPEI is a *derived drought index*, not a variable directly produced by climate models.
>
> In operational workflows (e.g., NOAA, BOM), agencies typically forecast precipitation and temperature first and then compute SPEI. Therefore, in the research literature on direct SPEI prediction, the standard non deep learning baselines are Persistence, ARIMA, and ETS. We have added all three in our evaluation, and the results are shown below:
>
> | Model       | MSE       | MAE       | R²        |
> | ----------- | --------- | --------- | --------- |
> | Persistence | 1.166     | 0.938     | −0.283    |
> | ETS         | 0.712     | 0.667     | 0.210     |
> | ARIMA       | 0.685     | 0.645     | 0.248     |
> | **RGMR**    | **0.318** | **0.423** | **0.651** |

---

### Note · Program_Chairs · 2026-01-17
**Submission Desk Rejected by Program Chairs**

The following references in this submission do not refer to real documents and/or have major errors in bibliographic information:

 Amir Shabani, Holger Fröhlich, and Michael Glaß. Scaleformer: Iterative multi-scale refining transformers for time series forecasting. In International Conference on Learning Representations (ICLR), 2023.
Sergio M. Vicente-Serrano et al. Understanding droughts: Models and trends. International Journal of Climatology, 35:180-190, 2015.